# Dual role of a GTPase conformational switch for membrane fusion by mitofusin ubiquitylation

Ramona Schuster[1], Vincent Anton[1], Tânia Simões[1], Selver Altin[1], Fabian den Brave[2], Thomas Hermanns[3], Manuela Hospenthal[4], David Komander[5,6,7], Gunnar Dittmar[8], R Jürgen Dohmen[3], Mafalda Escobar-Henriques[1]

**Mitochondria are essential organelles whose function is upheld by their dynamic nature. This plasticity is mediated by large dynamin-related GTPases, called mitofusins in the case of fusion between two mitochondrial outer membranes. Fusion requires ubiquitylation, attached to K398 in the yeast mitofusin Fzo1, occurring in atypical and conserved forms. Here, modelling located ubiquitylation to α4 of the GTPase domain, a critical helix in Ras-mediated events. Structure-driven analysis revealed a dual role of K398. First, it is required for GTP-dependent dynamic changes of α4. Indeed, mutations designed to restore the conformational switch, in the absence of K398, rescued wild-type-like ubiquitylation on Fzo1 and allowed fusion. Second, K398 is needed for Fzo1 recognition by the pro-fusion factors Cdc48 and Ubp2. Finally, the atypical ubiquitylation pattern is stringently required bilaterally on both involved mitochondria. In contrast, exchange of the conserved pattern with conventional ubiquitin chains was not sufficient for fusion. In sum, α4 lysines from both small and large GTPases could generally have an electrostatic function for membrane interaction, followed by posttranslational modifications, thus driving membrane fusion events.**

## Introduction

Mitochondria are dynamic organelles in eukaryotic cells, continuously remodelled by fusion and fission events (Tilokani et al, 2018). Mitochondrial morphology is directly linked to the metabolic state of the cell, and dysfunction in mitochondrial plasticity is associated with aging and disease (Schrepfer & Scorrano, 2016; Chen & Chan, 2017). The dynamic network of mitochondria is regulated by large dynamin-related GTPases, which undergo self-oligomerization and subsequent conformational changes thereby enabling membrane

mixing (Daumke & Praefcke, 2016). In addition to large GTPases, small GTPases such as Ras also play a major role in membrane remodelling events, being instead regulated by co-factors promoting nucleotide binding and hydrolysis (Peurois et al, 2019). The large GTPases involved in mitochondrial fusion are called mitofusins, Fzo1 in yeast and MFN1/2 in mammals (van der Bliek et al, 2013; Escobar-Henriques & Joaquim, 2019). Mutations in MFN2 cause the type 2A of Charcot–Marie–Tooth neuropathy, the most common degenerative disease of the peripheral nervous system (Zuchner et al, 2004; Barbullushi et al, 2019).

An increasing number of mitochondrial activities, including their fusion, depend on the post-translational modifier ubiquitin, a small cytosolic protein of 8.5 kD (Escobar-Henriques & Langer, 2014; D'Amico et al, 2017; Escobar-Henriques et al, 2019; Escobar-Henriques & Joaquim, 2019). Ubiquitin is a ubiquitous and highly conserved protein in eukaryotes which is covalently attached to lysine residues of its substrates. This affects substrate fate by either labelling them for degradation or altering their biochemical properties (Kwon & Ciechanover, 2017; Rape, 2018). The various effects of ubiquitylation are expanded by using the seven own internal lysine residues of ubiquitin, as well as its N terminus, to form homo- and heterotypic ubiquitin chains. Deubiquitylating enzymes (DUBs), ubiquitin-specific peptidases that remove ubiquitin from substrates, allow for an additional layer of regulation (Clague et al, 2019). Ubiquitylation of the yeast mitofusin Fzo1 is essential for its functionality in maintaining a healthy mitochondrial network (Anton et al, 2013). Moreover, the ubiquitin-specific AAA chaperone Cdc48/p97/VCP (Tanaka et al, 2010; Xu et al, 2011; Bodnar & Rapoport, 2017; Chowdhury et al, 2018; Simoes et al, 2018) was shown to be a major regulator of mitofusins (Tanaka et al, 2010; Xu et al, 2011; Bodnar & Rapoport, 2017; Chowdhury et al, 2018; Simoes et al, 2018). Mitofusins show an atypical and conserved ubiquitylation pattern (Fritz et al, 2003; Cohen et al, 2008; Ziviani et al, 2010; Rakovic et al, 2011; Simoes et al, 2018). Nevertheless, the relevance of this atypical pattern and how ubiquitin promotes outer membrane (OM) fusion are poorly understood.

[1]Institute for Genetics, Cologne Excellence Cluster on Cellular Stress Responses in Aging-Associated Diseases (CECAD), Center for Molecular Medicine Cologne, University of Cologne, Cologne, Germany   [2]Department of Molecular Cell Biology, Max Planck Institute of Biochemistry, Martinsried, Germany   [3]Institute for Genetics, University of Cologne, Cologne, Germany   [4]Institute of Molecular Biology and Biophysics, Eidgenössische Technische Hochschule Zürich, Zürich, Switzerland   [5]Medical Research Council Laboratory of Molecular Biology, Cambridge, UK   [6]Ubiquitin Signalling Division, The Walter and Eliza Hall Institute of Medical Research, Parkville, Australia   [7]Department of Medical Biology, The University of Melbourne, Melbourne, Australia   [8]Proteomics of Cellular Signalling, Luxembourg Institute of Health, Strassen, Luxembourg

Correspondence: Mafalda.Escobar@uni-koeln.de

Here, we investigate the properties of Fzo1 ubiquitylation on a structural level and their role for mitochondrial function. We reveal that Fzo1 ubiquitylation on helix 4 ($\alpha$4) of the GTPase domain is only permitted after a conformational switch within the helix, triggered by GTP hydrolysis. This is reminiscent of lysine-dependent remodelling events occurring in Ras GTPases and opens new perspectives for the functioning of large GTPases. Moreover, we show that the atypical ubiquitylation pattern of Fzo1 is stringently required for mitochondrial OM fusion and cannot be replaced by conventional ubiquitylation. In fact, the pattern has a function that goes beyond mitofusin stability. Instead, it allows for Fzo1 regulation by the deubiquitylase Ubp2 and by Cdc48/p97, two pro-fusion components.

# Results

## Unusual and conserved ubiquitylation pattern of mitofusins is linked to its fusion activity

Fusion requires ubiquitylation of Fzo1 (Anton et al, 2013). Moreover, the ubiquitylation pattern of mitofusins is highly conserved throughout the species (Cohen et al, 2008; Ziviani et al, 2010; Rakovic et al, 2011). It is not known, however, how the specific modifications resulting in this pattern promote mitochondrial fusion. The pattern consists of at least three distinct, unusually spaced bands, as resolved by SDS–PAGE and Western blot (Fig 1A, black arrows). The two heavier ubiquitylated bands correspond to K48-linked chains

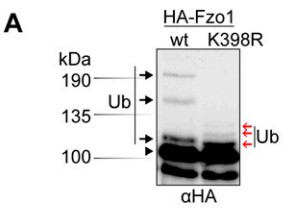

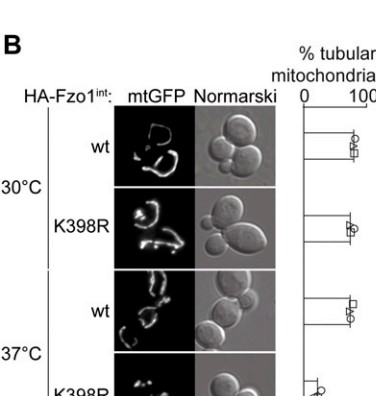

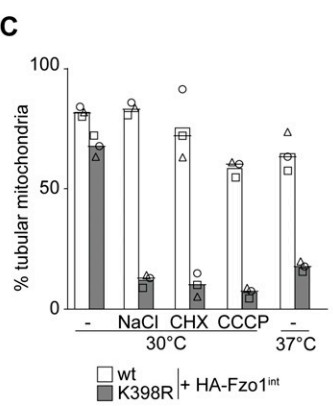

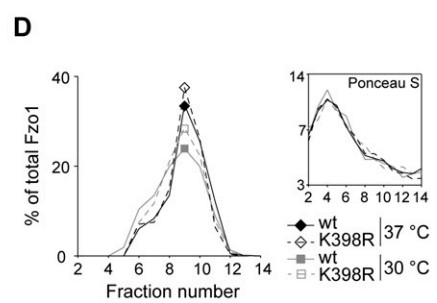

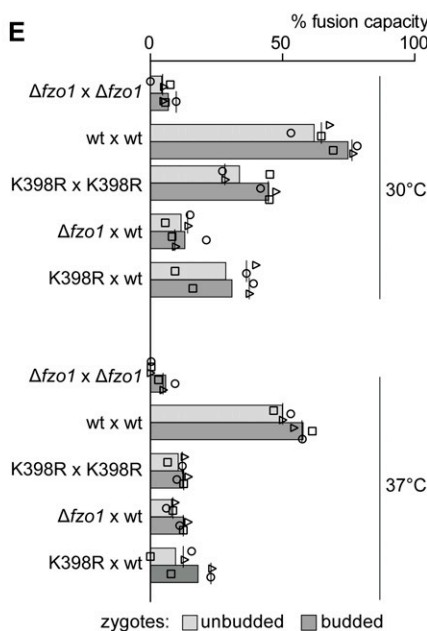

**Figure 1.  Conserved ubiquitylation pattern of Fzo1 is required on both fusion partners to drive efficient mitochondrial membrane fusion.**
**(A)** Fzo1 and Fzo1^K398R ubiquitylation pattern. Crude mitochondrial extracts from cells expressing HA-Fzo1 or HA-Fzo1$^{K398R}$ were solubilized, subjected to HA-immunoprecipitation and analysed by SDS–PAGE and Western blot using an HA-specific antibody. Unmodified and conserved ubiquitylated forms of Fzo1 are indicated by a black arrowhead or black arrows, respectively. K398R-specific ubiquitylation is indicated by red arrows. **(B)** Mitochondrial morphology of cells expressing genomically HA-tagged Fzo1 or Fzo1$^{K398R}$. Cells expressing HA-tagged Fzo1 (wt) or Fzo1$^{K398R}$ (K398R) were analysed for mitochondrial tubulation after expressing a mitochondrial-targeted GFP plasmid. Cellular (Nomarski) and mitochondrial (mtGFP) morphology were visualized by fluorescence microscopy. Right panel: quantification of three independent experiments (with more than 200 cells each), depicting the percentage of tubular mitochondria. Individual relative values, mean, and median are indicated by geometric symbols, bar, and horizontal line, respectively. Scale bar; 5 $\mu$m. **(C)** Mitochondrial morphology of cells expressing HA-Fzo1 and HA-Fzo1$^{K398R}$ upon cellular stresses. Cells expressing genomically integrated HA-Fzo1 and HA-Fzo1$^{K398R}$ were grown to the exponential growth phase at 30°C. Cultures were then divided and further grown for 1 h at 37°C or at 30°C without any treatment or with high salt (0.5 M NaCl), sublethal doses of cycloheximide (1.7 $\mu$M CHX) or CCCP for mitochondrial membrane uncoupling (10 $\mu$M). Mitochondrial morphology was analysed as in Fig 1B. The quantification of three independent experiments shows the mean (bar), the median (line), and the individual values (geometric symbols) in % tubular mitochondria. **(D)** Complex formation of HA-tagged Fzo1 or Fzo1$^{K398R}$. Sucrose gradient centrifugation was performed with solubilized crude mitochondrial extracts of strains genomically expressing HA-tagged Fzo1 or Fzo1$^{K398R}$ grown at 30°C and 37°C. Gradients were fractionated, proteins were precipitated with trichloroacetic acid, and the samples were analysed by SDS–PAGE and Western blot using an HA-specific antibody. HA signals and Ponceau S staining (inset: used as a gradient fractionation control) were quantified. **(E)** In vivo fusion assay. Early mating stages (unbudded zygotes) and late mating stages (budded zygotes) are scored. Mating of Δ*fzo1* cells (a or $\alpha$) expressing 3xMyc-Fzo1 or 3xMyc-Fzo1$^{K398R}$ and mtGFP or mtRFP, each under the control of an inducible *GAL1* promoter. The cells were grown in SC supplemented with 2% raffinose to the exponential growth phase. 2% galactose was added for 1 h to induce Fzo1 expression. 2% glucose was then added to stop Fzo1 expression. The cells were mixed 1 h after glucose addition, and fluorescence microscopy pictures were taken after an additional 8 h. The mixing of fluorophores within the mitochondria was quantified. Homotypic and heterotypic mating was performed, at 30°C and 37°C, and mixing of mitochondria was observed in unbudded (light grey) or budded (dark grey) zygotes. The quantification of three independent experiments (30 or more mating events each) shows the mean (bar), the median (line), and the individual values (geometric symbols) in % fusion capacity. Ub, ubiquitin.

conjugated to lysine 398 in Fzo1 (Anton et al, 2013). However, the K398R mutant still displays higher molecular weight forms, which are reminiscent of the canonical pattern of a ubiquitin ladder (Fig 1A, right lane, red arrows). Expression of Myc-tagged ubiquitin as the sole ubiquitin species proved the presence of ubiquitin on Fzo1$^{K398R}$ by inducing shifts in the electrophoretic mobility of these modified forms (Fig S1A). Furthermore, as it is the case for wild-type (wt) Fzo1, ubiquitylation of Fzo1$^{K398R}$ was impaired by preventing GTP binding (Fzo1$^{D195A}$) and hydrolysis (Fzo1$^{T221A}$) (Fig S1B). Nevertheless, cells expressing Fzo1$^{K398R}$ were shown to have reduced respiratory capacity (Anton et al, 2013), suggesting that the conserved ubiquitylation pattern was required for full Fzo1 functionality in promoting mitochondrial fusion. Thus, mitochondrial morphology was visualized by expression of GFP targeted to mitochondria. Strikingly, a severe tubulation defect could be observed in Fzo1$^{K398R}$ cells grown at 37°C (Fig 1B), revealing that Fzo1$^{K398R}$ behaves like a thermo-sensitive allele. Likewise, other stresses, such as high salt, sub-lethal doses of cycloheximide or mitochondrial membrane potential impairment resulted in defective mitochondrial tubulation (Fig 1C). These results indicate that upon cellular stress, mitochondria do not fuse efficiently in the absence of the conserved pattern.

### No other ubiquitin-like modifier (UBL) accounts for the large molecular shift in Fzo1 ubiquitylation

Given the importance of the conserved pattern for mitochondrial tubulation, we were prompted to analyse it in more detail. First, in vitro deubiquitylation assays were performed (Hospenthal et al, 2015) on HA-Fzo1 immobilized on HA-coupled beads. Subsequent treatment with purified DUB USP21 efficiently cleaved all ubiquitylated forms of Fzo1 (Fig S2A). Moreover, kinetic experiments using suboptimal DUB concentrations indicated a similar sensitivity profile of the different higher molecular weight forms of Fzo1 toward USP21 cleavage (Fig S2B). Given that ubiquitin itself is only an 8.5-kD protein, we investigated what leads to the observed large molecular shift between the different ubiquitylated forms of Fzo1. To address this problem, we tested whether other UBLs would be conjugated to ubiquitylated Fzo1. However, deletion of *ATG12*, *HUB1*, *URM1*, *RUB1*, or *ATG8* did not change the pattern of Fzo1 ubiquitylation (Fig S2C). Similarly, no shift in the ubiquitin forms was observed upon impairment of the SUMO E1 enzyme (*uba2$^{ts}$* mutant strains) (Fig S2D). We noticed that deletion or mutation of the other UBLs generally increased the ubiquitylation of Fzo1, likely reflecting cellular compensatory effects. Furthermore, MFN2 was suggested to partially localize to the ER (de Brito & Scorrano, 2008). Therefore, we tested if the abnormal shift in Fzo1 ubiquitylation could reflect additional modification of Fzo1 by glycosylation, which occurs at the ER. However, glycosidase treatment of mitochondrial extracts showed no change in the pattern of Fzo1 ubiquitylation, suggesting that glycosylation is not involved in the mass-shift (Fig S2E). In contrast and as expected, Psd1 clearly shifted upon glycosidase treatment (Friedman et al, 2018) (Fig S2E). Finally, we considered the possibility that ubiquitin attached to Fzo1, rather than Fzo1 itself, could be modified. To test this, DUB treatments were performed in cells exclusively expressing Myc-tagged ubiquitin and the electrophoretic running behaviour of Myc-ubiquitin was observed.

However, Myc-ubiquitin released from Fzo1 upon DUB treatment migrated at the expected height according to its molecular weight, similarly to Fzo1$^{K398R}$, which presents the canonical ladder-like pattern (Fig S2F). In conclusion, we were not able to identify other modifications on Fzo1 responsible for the conserved ubiquitylation pattern. However, we noticed changes in the apparent molecular weight shift between the different ubiquitylated forms, when using variations in SDS–PAGE conditions, namely, different amounts of bis-acrylamide and running chambers (Fig S2G). Depending on these variations, and taking the different running behaviours of the protein marker into account, Fzo1 ubiquitylated forms appear to have the apparent sizes 115, 160, and 200 kD for 0.8% bis-acrylamide using the HOEFER system or 120, 175, and 215 kD for 0.2% bis-acrylamide using the Bio-Rad system. In fact, addition of ubiquitin chains to a substrate means that this protein is no longer composed of a linear amino acid chain, thus explaining possible running artefacts, even in the presence of SDS. In sum, our results suggest that instead of the presence of other post-translational modifications, the conserved uncommon running behaviour reflects conformations in mitofusins resistant to complete denaturation by SDS–PAGE.

### Atypical Fzo1 ubiquitylation is required on both fusing partners

Mitochondrial fusion is a stepwise process that depends on oligomerization of mitofusins (Ishihara et al, 2004; Anton et al, 2011; Cohen et al, 2011; Engelhart & Hoppins, 2019; Sloat et al, 2019). First, GTP binding allows formation of Fzo1 dimers, which further oligomerize *in trans*, that is, upon getting in contact with dimers of opposing mitochondrial membranes (Anton et al, 2011). To observe Fzo1 complex formation in vitro, solubilized mitochondria can be analysed for *trans* assembly by sucrose gradient centrifugation, as previously described (Anton et al, 2011). In this experimental setup, mitochondria and, thus, Fzo1 molecules have to be in close contact to each other to be able to form higher oligomers. At low mitochondrial and Fzo1 density, Fzo1 will only form dimers (Fig S3A, "low"). If the density of mitochondria and Fzo1 in the reaction is 10-fold higher, Fzo1 forms higher oligomers (Fig S3A, "high"). Given that the K398R mutant variant of Fzo1 was not able to mediate mitochondrial fusion at 37°C, we analysed if the complex formation of Fzo1 was disturbed. However, cells expressing the K398R mutant of Fzo1 displayed the same ability as wt Fzo1 to form oligomeric complexes, both at 30°C and 37°C (Figs 1D and S3B). This indicated that the conserved ubiquitylation pattern of Fzo1 is required at a later stage of the fusion process, downstream of Fzo1 assembly *in trans*.

Next, we analysed if wt Fzo1 is required on both fusing membranes. To this aim, we directly quantified the fusion capacity of Fzo1 and Fzo1$^{K398R}$, using an assay previously described (Nunnari et al, 1997). Briefly, mitochondrial targeted GFP (mtGFP) or red fluorescent protein (mtRFP) are separately expressed in cells of each mating type. Fusion events are identified by co-localization of the two fluorophores after mating. To tightly control the exclusive presence of only one Fzo1 variant and one fluorescent marker in each mating type, their expression was shut off before mating, by using the glucose repressible *GAL1* promoter (Anton et al, 2019). We first confirmed that Fzo1 was still present after synthesis shutoff (Fig S3C). The adapted mating assay demonstrated, as expected,

mitochondrial content mixing between two wt strains (wt × wt) indicating mitochondrial fusion (Fig 1E). Fusion was only slightly affected by increasing the temperature from 30°C to 37°C. In contrast, absence of *FZO1* prevented co-localization of the two fluorophores, at 30°C or 37°C, both shortly after mating ("unbudded") and after the zygotes budded ("budded") (Fig 1E, Δ*fzo1* × Δ*fzo1*). Consistent with the mitochondrial morphology defects (Fig 1B), fusion capacity was extremely compromised when cells expressed Fzo1[K398R] (K398R × K398R), especially at 37°C. Moreover, this was also the case for heterotypic fusion (K398R × wt). Taken together, our results show that efficient mitochondrial fusion relies on the presence of the conserved ubiquitylation pattern of Fzo1, on both fusion partners.

### Canonical ladder-like ubiquitylation destabilizes Fzo1 and possesses no fusogenic activity

The Fzo1[K398R] variant displays a ubiquitylation pattern typical for unstable proteins, known to be proteasomal targets. In fact, we could previously show that the K398R mutation affects Fzo1 stability, by blocking protein synthesis with cycloheximide (CHX) and chasing the pre-existing protein over a course of 3 h (Anton et al, 2013 and Fig 2A). To test whether this is due to proteasomal degradation of Fzo1, MG132 was added 1 h before the CHX treatment to inhibit the proteasome. Indeed, this slowed down the turnover of Fzo1[K398R], suggesting that it is a proteasomal target (Fig 2A). Next, we reduced the formation of K48-linked ubiquitin chains, generally targeting proteins for proteasomal turnover, by ectopic expression of K48R-mutated ubiquitin. This led to increased levels of unmodified Fzo1[K398R], consistent with loss of the degradation signal (Fig S4A). Concomitantly, an increased amount of mono-ubiquitylated Fzo1 could be observed, together with impaired fusion capacity (Fig S4A and B). Of note, as K48 is essential for cellular viability, Ub[K48R] was expressed on top of endogenous wt ubiquitin, explaining the remaining ubiquitin chains in both Fzo1 and Fzo1[K398R]. Together, these results confirm that the canonical ubiquitylation pattern present in Fzo1[K398R] address it for proteasomal degradation.

Lower protein levels of Fzo1 could be causative of the impaired fusion capacity of Fzo1[K398R]. Therefore, we ectopically expressed Fzo1[K398R] from a centromeric plasmid, in addition to genomically encoded Fzo1[K398R], which rescued Fzo1[K398R] back to wt-like levels (Fig 2B). However, this did not rescue mitochondrial tubulation at 37°C, indicating that the observed mitochondrial fusion defect is not due to the low levels of the Fzo1[K398R] protein (Fig 2C). Similarly, restoration of Fzo1 steady state levels by inhibiting Fzo1 turnover with MG132 (Fig S4C) did not rescue mitochondrial tubulation (Fig S4D) despite increased ubiquitylation of Fzo1[K398R] (Fig S4E). This confirms that the ubiquitylated forms of Fzo1[K398R] harbour no fusogenic activity at 37°C. In sum, our results indicate that the functional impairment of the Fzo1[K398R] mutant is not due to reduced levels of Fzo1.

### Loss of wild-type ubiquitylation renders Fzo1 insensitive to the DUB Ubp2 and the AAA-ATPase Cdc48

We previously observed that the DUB Ubp2 removes degradative ubiquitin chains from wt Fzo1, protecting it from proteasomal turnover (Anton et al, 2013). Indeed, both wt Fzo1 in Δ*ubp2* cells and

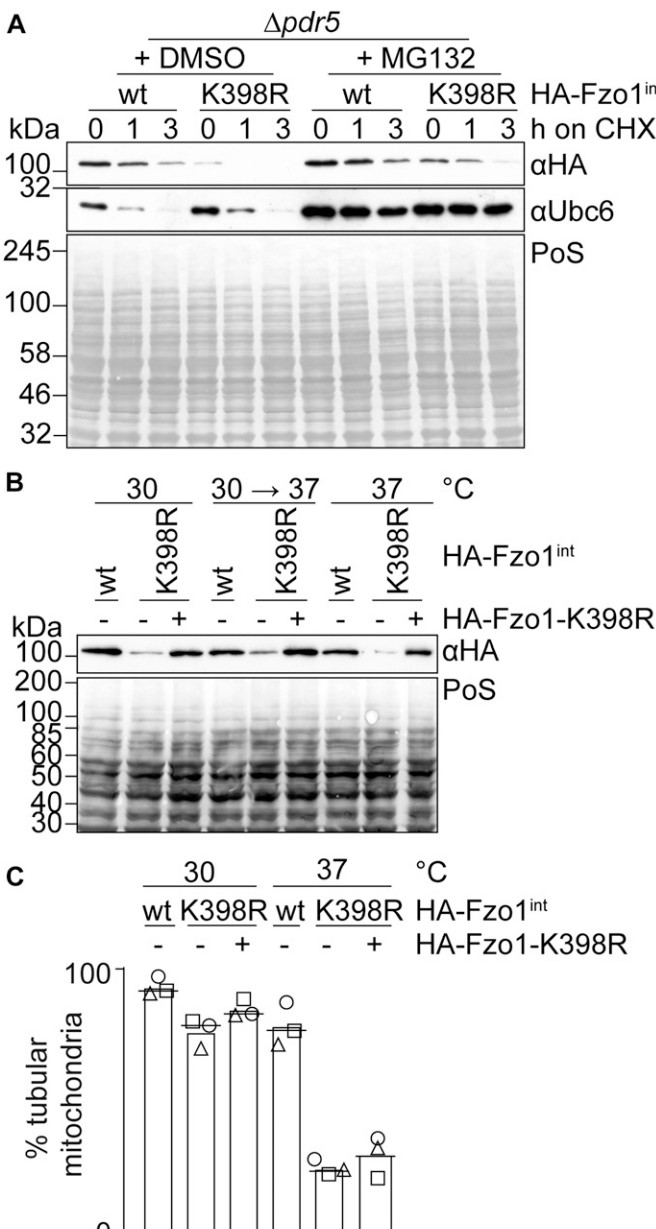

**Figure 2. Rescue of Fzo1 protein levels is not sufficient to restore mitochondrial fusion.**
**(A)** Stability and proteasome dependence of endogenously HA-tagged Fzo1 and Fzo1[K398R]. Exponentially growing cells defective for the multidrug exporter Pdr5 (Δ*pdr5*), expressing endogenously HA-tagged Fzo1 (wt) or Fzo1[K398R] (K398R), were treated with proteasomal inhibitor MG132 (or DMSO as a control) for 1 h before administration of the translation inhibitor cycloheximide (CHX). Samples were taken 0, 1, or 3 h after protein synthesis was shut off with CHX. Total cellular extracts were prepared and analysed by SDS–PAGE and Western blot using HA-specific and Ubc6-specific (as an unstable protein control) antibodies. **(B)** Rescue of HA-Fzo1[K398R] steady state levels by ectopic expression of HA-Fzo1[K398R]. HA-Fzo1[K398R] (or the corresponding vector control) was ectopically expressed using a centromeric plasmid in cells already expressing genomically HA-tagged Fzo1[K398R], which were grown at 30°C, 30°C with a subsequent shift to 37°C for 3 h or 37°C, as indicated. Total cellular extracts were analysed by SDS–PAGE and Western blot using an HA-specific antibody. **(C)** Mitochondrial morphology of cells expressing equal levels of HA-tagged Fzo1 and Fzo1[K398R]. **(B)** Mitochondrial morphology was analysed as in Fig 1B using cells as described in (B). PoS, Ponceau S.

Fzo1[K398R] in wt cells responded equally to MG132 treatment (Fig 3A). Therefore, the lower protein levels and proteasomal sensitivity of Fzo1[K398R] were reminiscent of wt Fzo1 when expressed in the absence of Ubp2. This prompted us to investigate if Fzo1[K398R] is protected from turnover by Ubp2, by testing whether the absence of the Ubp2 protein affected the stability of Fzo1[K398R]. Interestingly, CHX chases showed that Δubp2 and Fzo1[K398R] cells have similar turnover kinetics of Fzo1 (Fig 3B) even in the presence of additional Fzo1[K398R] copies to rescue wt-like levels (Fig S5A). Moreover, deletion of UBP2 in cells expressing Fzo1[K398R] had no additive defect (Fig 3B). Interestingly, in contrast to our observations for Fzo1[K398R] (Fig S4D), proteasome inhibition in Δubp2 cells allowed rescuing mitochondrial network formation (Anton et al, 2013). This difference is likely due to the fact that Δubp2 cells maintain the wt-like fusogenic ubiquitylation pattern on Fzo1 (Fig 3C). In conclusion, Fzo1[K398R] is not stabilized by Ubp2, indicating that the proteolytic ubiquitin forms from Fzo1[K398R] are not the same that Ubp2 recognizes on a wt protein.

In addition to Ubp2, the ubiquitin chaperone Cdc48 was also identified as a pro-fusion regulator of Fzo1 (Chowdhury et al, 2018;

Simoes et al, 2018). Cdc48 interacts with Fzo1 and prevents its proteasomal turnover. Therefore, we first asked whether Cdc48 was able to interact with Fzo1[K398R]. To test this, we used cells expressing either endogenously HA-tagged Fzo1, or endogenously HA-tagged Fzo1[K398R], together with HA-Fzo1[K398R] from a centromeric plasmid, to compensate for the different levels of Fzo1 and Fzo1[K398R]. Co-immunoprecipitation experiments confirmed the physical interaction between Cdc48 and Fzo1 but showed impaired binding of Cdc48 to Fzo1[K398R] (Fig 3D). In contrast, Cdc48 bound Fzo1 in Δubp2 cells, consistent with the presence of wt ubiquitylation. Furthermore, the capacity of Cdc48 to protect Fzo1[K398R] from proteasomal turnover was tested by comparing its steady state levels and stability in the presence of wt Cdc48 or the hypomorphic mutant cdc48-2. Consistent with impaired recognition of Fzo1[K398R] by Cdc48, we could observe that the protein levels and stability of this Fzo1 variant was nearly insensitive to the absence of functional Cdc48 (Figs 3E and S5B). Together, these results reinforce the importance of the conserved pattern in responding to the known pro-fusion regulators of Fzo1 and of mitochondrial fusion.

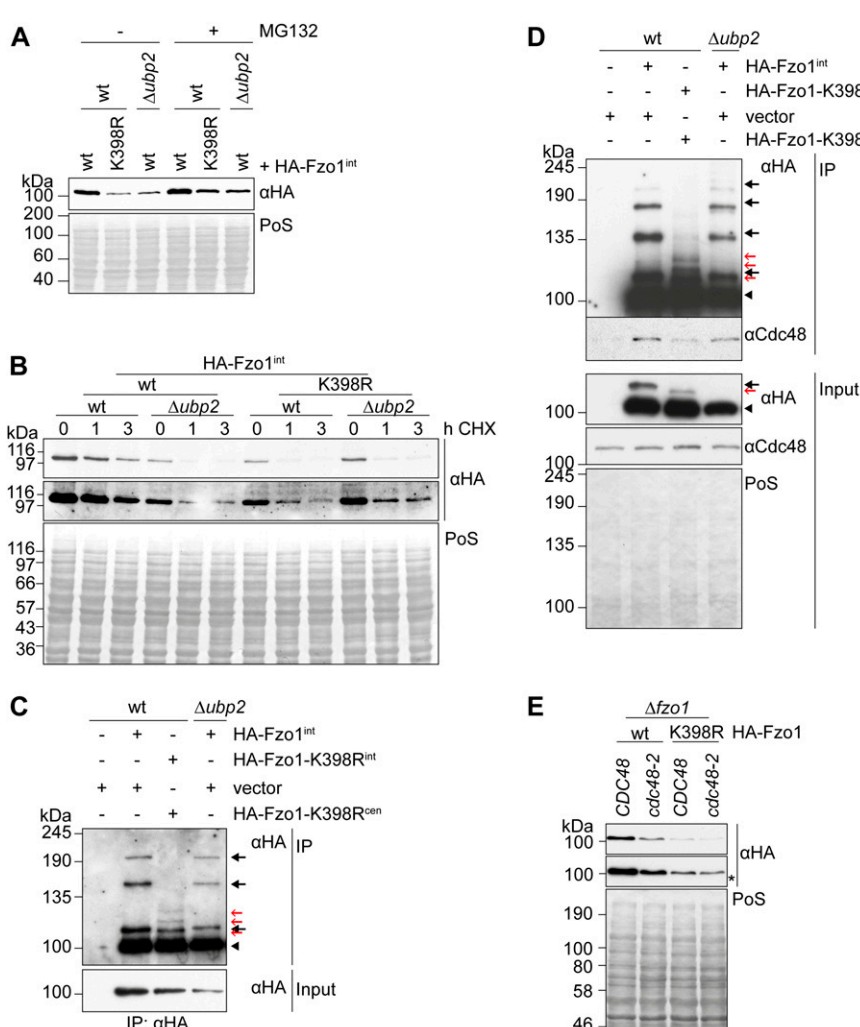

**Figure 3. Loss of wild-type ubiquitylation renders Fzo1 insensitive to Ubp2 and Cdc48.**
**(A)** Effect of proteasome inhibition and *UBP2* deletion in the steady state levels of Fzo1 and Fzo1[K398R]. The proteasome was inhibited as described in Fig 2A in Δp*dr5* (indicated as wt) or in Δp*dr5* Δ*ubp2* (indicated as Δ*ubp2*) cells, expressing genomically integrated HA-tagged Fzo1 or Fzo1[K398R]. Total cellular extracts were analysed by SDS–PAGE and Western blot using an HA-specific antibody. **(B)** Stability of genomically HA-tagged Fzo1 or Fzo1[K398R] in wt or Δ*ubp2* cells. The turnover of genomically HA-tagged Fzo1 (wt) or Fzo1[K398R] (wt) in wt or Δ*ubp2* cells was assessed as in Fig 2A. The samples were analysed by SDS–PAGE and Western blot using an antibody against HA. **(C)** Comparison of the ubiquitylation pattern of Fzo1 upon *UBP2* deletion or Fzo1-K398R mutation. Crude mitochondrial extracts from wt cells (expressing genomically HA-tagged Fzo1 or Fzo1[K398R]) or Δ*ubp2* cells (expressing genomically HA-tagged Fzo1) were solubilized, subjected to HA-immunoprecipitation, and analysed by SDS–PAGE and Western blot using an HA-specific antibody. Forms of Fzo1 are indicated as in Fig 1A. **(D)** Analysis of Cdc48 immunoprecipitation with genomically HA-tagged Fzo1 or Fzo1[K398R] in wt or Δ*ubp2* cells. **(C)** Crude mitochondrial extracts from cells as in (C) (but grown at 37°C) were solubilized, subjected to co-immunoprecipitation, and analysed by SDS–PAGE and Western blot using HA- and Cdc48-specific antibodies. Forms of Fzo1 are indicated as in Fig 1A. **(E)** Steady state levels of HA-tagged Fzo1 and Fzo[K398R] in wt and *cdc48-2* cells. Total cellular extracts of wt or *cdc48-2* cells, ectopically expressing HA-tagged Fzo1 or Fzo1[K398R], were analysed by SDS–PAGE and Western blot, using an HA-specific antibody. IP, immunoprecipitation; PoS, Ponceau S.

## Conformational readjustment in α4 of Fzo1 can compensate for the ubiquitylation defect in Fzo1^K398R

Our observations suggest that ubiquitylation of mitofusins locks these GTPases in stable and conserved conformations. To further investigate the role of K398 in the conformation of the GTPase domain and in the fusion functions of Fzo1, we modelled it to the crystal structures of the minimal GTPase domain (MGD) of MFN1 (Qi et al, 2016; Cao et al, 2017; Yan et al, 2018) (Fig 4A). Fzo1^K398R presents ubiquitylated forms, meaning that a residue other than K398 is ubiquitylated. Therefore, we aimed to identify additional ubiquitin target sites on the protein. We first investigated eight lysine residues proximal to K398 and located to the surface of the Fzo1 protein, that is, accessible for post-translational modifications (Fig 4A, annotated in orange). We used the low steady state levels of HA-Fzo1^K398R as a readout, assuming that the absence of the Fzo1^K398R-specific ubiquitylation would stabilize the protein. Thus, we screened which of those lysines, when mutated to arginine in a K398R background, presented increased steady state levels of Fzo1. We observed a partial rescue of the steady state levels of Fzo1 in the variant K398,382R (Fig S6A). This appeared to be specific because no stabilization of Fzo1 could be observed when testing other lysine residues. Indeed, mutations additional to K398R in three highly conserved lysines, or in nine lysine residues found to be ubiquitylated by mass spectrometry, did not increase the levels of Fzo1 (Fig S6B and C

and Table S1). Next, we investigated the effect on ubiquitylation of this K398,382R variant of Fzo1. Strikingly, albeit at a lower level than wt, the double mutation K398,382R restored the ubiquitylation pattern of Fzo1 (Figs 4B and S6D). Moreover, although Fzo1^K398,382R rescued Fzo1 ubiquitylation, a K382R single mutant showed no differences to wt Fzo1 (Fig S6D). Once again, this effect was specific to K382 because the other mutants tested did not re-gain wt-like ubiquitylation (Fig S6B and C). Consistently, the K398,382R variant also restored interaction with Cdc48 (Fig 4B). This supports the conclusion that the K398R variant is turned over more rapidly because of a defective interaction with Cdc48. We then tested if the double mutant of K398R and K382R, displaying the conserved ubiquitylation pattern of Fzo1, was able to rescue mitochondrial fusion. To this, we scored mitochondrial tubulation by fluorescence microscopy. Importantly, we found that the double mutant could rescue the fusion defect of Fzo1^K398R at 37°C to nearly wt fusion levels (Fig 4C). As expected, there was no morphology anomaly observable at 30°C (Fig S6E).

Interestingly, K398 and K382 are both located in the conserved α4 within the GTPase domain (Fig 4A, zoom-in). We predicted that the K398R mutation could create disturbed conformations in α4, perhaps rescued upon further mutation of K382. Therefore, we modelled MGD-Fzo1, MGD-Fzo1^K398R, and MGD-Fzo1^K398,382R on MGD-MFN1 bound to GDP (Cao et al, 2017). We chose the post-hydrolysis state, given that Fzo1 ubiquitylation occurs after GTP hydrolysis (Anton

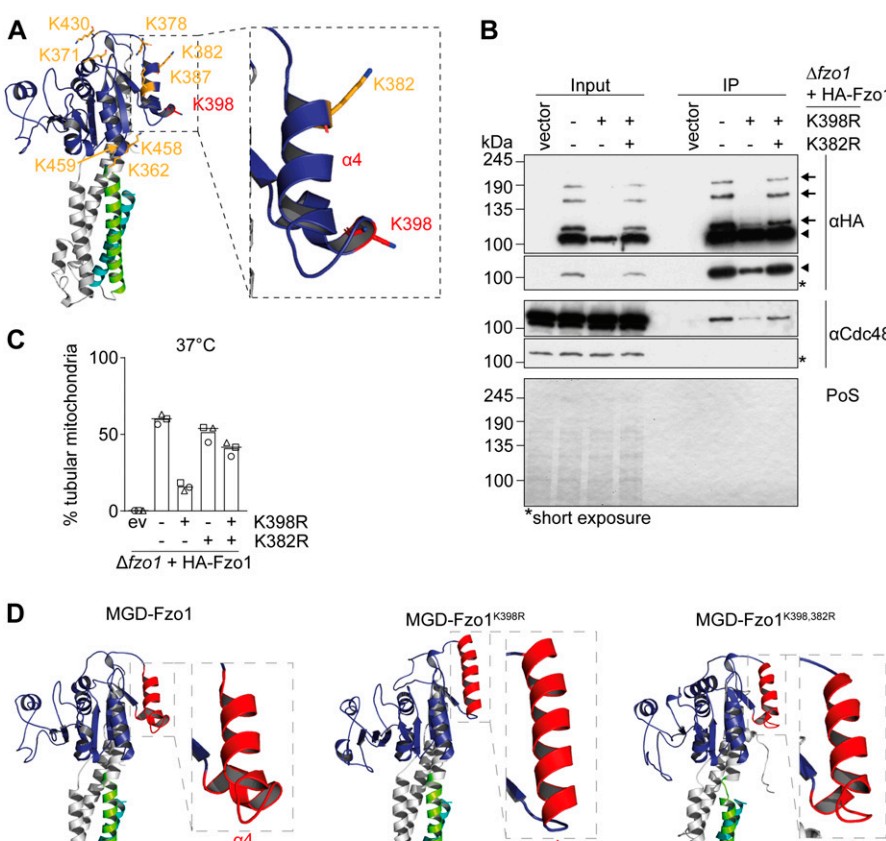

**Figure 4. Off-site mutation restores Fzo1 ubiquitylation and tubulation in Fzo1^K398R mutants.**
**(A)** Surface lysines annotated on a model of Fzo1. A structural model of the Fzo1-MGD (aa 61–491-linker-826-856) modelled on MFN1-MGD bound to GDP-AlF_4^- (PDB ID: 5GOM, c-score: −1.24) (Cao et al, 2017). The c-score ranges from −5 to +2 where a more positive score reflects a model of better quality. Surface lysines on Fzo1-MGD are annotated in orange and lysine 398 is annotated in red. The GTPase domain is shown in blue, HR1 is shown in green, and HR2 is shown in cyan. The inset shows a zoom-in of the helix α4 of the GTPase domain. **(B)** Analysis of Fzo1^K398,382R ubiquitylation and co-immunoprecipitation with Cdc48. Crude mitochondrial extracts from Δfzo1 cells expressing HA-tagged Fzo1, Fzo1^K398R, and Fzo1^K398,382R were solubilized, subjected to immunoprecipitation, and analysed by SDS–PAGE and Western blot using HA- and Cdc48-specific antibodies. Forms of Fzo1 are indicated as in Fig 1A. **(C)** Mitochondrial morphology of cells expressing HA-tagged Fzo1 mutated for K398R and/or K382R. Mitochondrial morphology of cells expressing wt or mutants of HA-Fzo1, as indicated, and grown at 37°C was analysed as in Fig 1B. **(D)** Structural models of Fzo1. **(A)** Wt, K398R, and K398,382R Fzo1 was modelled on MFN1-MGD bound to GDP-AlF_4^- (PDB ID: 5GOM, c-score: −1.24, −2.36 and −2.52, respectively), as in (A). The inset shows a zoom of the helix α4 of the GTPase domain, in red. PoS, Ponceau S.

et al, 2011). As we predicted, α4 in wt and K398,382R Fzo1 looked very similar, being interrupted by a short loop, in contrast to the long, continuous helix present in the K398R Fzo1 variant (Fig 4D). Therefore, we suggest that the preference of a lysine residue at position 398 plays a crucial role for the conformation of α4, required for mitochondrial fusion.

### Off-site rescue of Fzo1 ubiquitylation is required for functionality

Our results demonstrate that in the absence of K398, additional mutation of K382 allows for the conserved ubiquitylation pattern to be formed on yet another lysine residue. Given that lysine K387 is located in α4 (Fig 5A), in between K398 and K382, we predicted this lysine would be ubiquitylated in the Fzo1$^{K398,382R}$ variant. To substantiate the importance of ubiquitylation in α4, we mutated this residue on top of the double mutant. Consistently, in the triple mutant Fzo1$^{K398,382,387R}$, the restored conserved ubiquitylation pattern and the physical interaction with Cdc48 were lost (Fig 5B and C). Furthermore, as expected, the triple mutant Fzo1$^{K398,382,387R}$ could no longer mediate mitochondrial tubulation at 37°C (Fig 5D), whereas no fusion defects could be detected at permissive temperature (Fig S7A). Thus, short ubiquitin chains on Fzo1 that are normally attached to K398 can also be conjugated to K387 in the absence of K398, provided that the conformation of α4 can be rescued by additionally mutating K382 to arginine.

Importantly, it was suggested that MFN1 changes from a closed to an open dimer upon GTP hydrolysis (Yan et al, 2018), based upon

structures obtained in the presence of GDP-BeF$_3^-$ and GDP-AlF$_4^-$, respectively (Fig 5E). Therefore, we analysed how the nucleotide state affected the conformation of α4 in Fzo1 (Fig 5E, annotated in red). Strikingly, wt Fzo1 recapitulated the conformational change from a long α4 to a shorter and interrupted one, driven by GTP hydrolysis (Fig 5E). Moreover, α4 of the wt Fzo1 in the pre-hydrolysis state (Fig 5E) resembled the long uninterrupted helix observed in the post-hydrolysis state of Fzo1$^{K398R}$. This supports that the K398R mutation blocks fusion by altering the conformation of α4, preventing it from adopting the interrupted conformation seen after GTP hydrolysis. Consistently, the double mutant K398,382R, which restored the wt-like kink, restores ubiquitylation and fusion.

In summary, we identified a critical role of α4 in Fzo1 allowing the conserved ubiquitylation pattern and thus allowing Cdc48 interaction and enabling mitochondrial fusion. The regulation of mitofusin conformations has been recently identified as a target for drug treatment (Rocha et al, 2018). This underlines the far-reaching importance of understanding how conformational changes impact on mitofusins.

## Discussion

Mitochondrial fusion is a tightly controlled process, essential for the maintenance of cellular respiration and governed by ubiquitylation of mitofusins. However, their mechanism of action remains elusive.

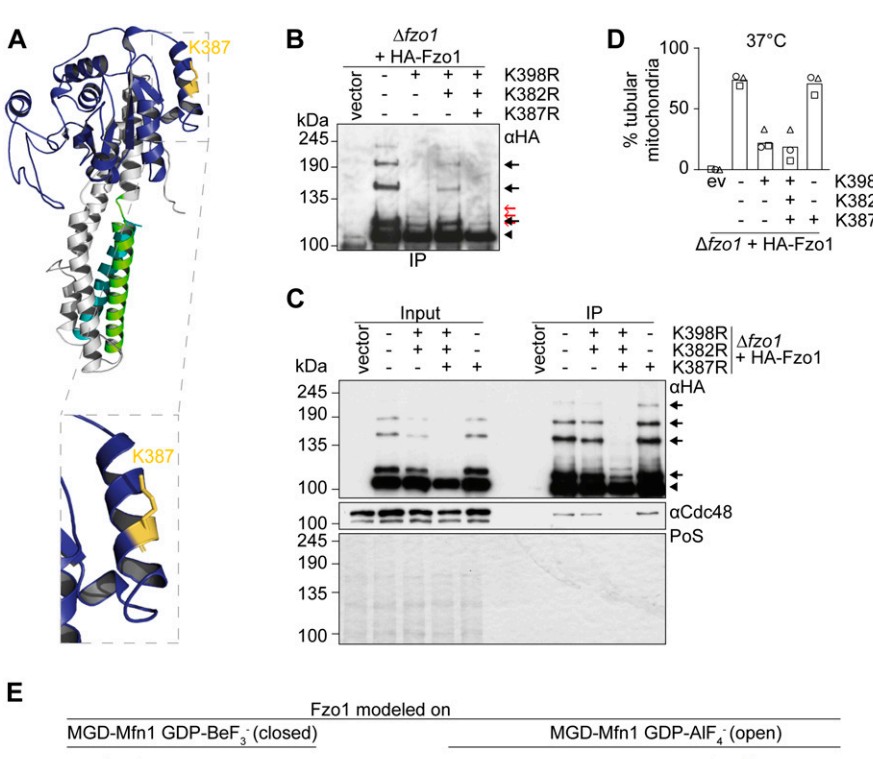

**Figure 5. Combined lysine mutations in Fzo1 reveal an important role of the GTPase α4.**
**(A)** MGD-Fzo1$^{K398,382R}$ was modelled on MGD-MFN1 as described in Fig 4A (Cscore –2,52). Residue 387 is annotated in yellow. The inset shows a zoom-in of α4 of the GTPase domain. **(B)** Analysis of Fzo1 ubiquitylation of cells expressing HA-tagged Fzo1 mutated for K398R and/or K382R and/or K387R. Crude mitochondrial extracts of cells expressing wt or mutants of HA-Fzo1, as indicated, were solubilized, subjected to immunoprecipitation, and analysed by SDS–PAGE and Western blot, using an HA-specific antibody. Forms of Fzo1 are indicated as in Fig 1A. **(C)** Analysis of Cdc48 co-immunoprecipitation with HA-tagged Fzo1, Fzo1$^{K398,382R}$, Fzo1$^{K398,382,387R}$, or Fzo1$^{K387R}$. Crude mitochondrial extracts from Δfzo1 cells expressing the indicated mutants of HA-tagged Fzo1 were solubilized, subjected to immunoprecipitation, and analysed by SDS–PAGE and Western blot using HA- and Cdc48-specific antibodies. Forms of Fzo1 are indicated as in Fig 1A. **(D)** Mitochondrial morphology of cells expressing HA-tagged Fzo1 mutated for K398R and/or K382R and/or K387. Mitochondrial morphology of cells expressing wt or mutants of HA-Fzo1, as indicated, and grown at 37°C was analysed as in Fig 1B. **(E)** Structural model of Fzo1 modelled on MFN1-MGD. Fzo1-MGD was modelled on MFN1-MGD bound to GDP-BeF$_3^-$ (centre left, PDB ID: 5YEW, c-score –1.17) (Yan et al, 2018) or GDP-AlF$_4^-$ (centre right, PDB ID: 5GOM, c-score –1.24) (Cao et al, 2017). Structures are depicted as closed and open dimers, indicating prehydrolysis and post-hydrolysis states, respectively. PoS, Ponceau S.

Mitofusins share a conserved ubiquitylation pattern, from yeast (Fzo1) to mammals (MFN1/MFN2) (Cohen et al, 2008; Ziviani et al, 2010; Rakovic et al, 2011; Simoes et al, 2018). Here, we show that Fzo1 ubiquitylation per se or modulation of its expression levels are not sufficient for efficient mitochondrial fusion. Instead, the conserved ubiquitylation pattern of Fzo1 is required and needs to be present on both mitochondrial fusion partners. Using structure modelling, we now pinpoint the ubiquitylation of Fzo1 to a functional location on the Fzo1 molecule and further dissect the fusion process. We show that Fzo1 is ubiquitylated in the conserved α4 of the GTPase domain. α4 structural modelling indicates that α4 undergoes conformational changes driven by GTP hydrolysis, permitting sub-sequent ubiquitylation.

### The atypical ubiquitylation pattern on mitofusins

We reveal new insights into the regulation and the molecular properties of ubiquitylated Fzo1. First, mutation of lysine 398, a residue required for the conserved ubiquitylation pattern, leads to a ladder-like ubiquitylation pattern, which renders Fzo1 a highly unstable protein incapable of efficiently promoting mitochondrial fusion upon cellular stress. However, the lack of fusion efficiency is not a result of decreased amounts of Fzo1 in the mutant, as the rescue to wt-like levels of Fzo1$^{K398R}$ did not restore mitochondrial tubulation. Second, the ubiquitylation of mitofusins show unusual electrophoretic shifts between the different forms. Nevertheless, we could exclude the involvement of the other known UBLs in this size shift. Furthermore, cleavage by DUBs indicated that the abnormal shift was not a result of other modifications on ubiquitin itself. This suggests that, instead, ubiquitylation in K398 could induce stable and conserved conformations of Fzo1, resulting in the electrophoretic shift observed by Western blot. Consistently, several studies showed that ubiquitylation of proteins can change their conformation, even in denaturing SDS gels (Varshavsky, 1995; Sagar et al, 2007; Tsutakawa et al, 2011; Zhang et al, 2012; Morimoto et al, 2016).

### Structural rearrangements of Fzo1 allow its ubiquitylation and mitochondrial OM fusion

The main ubiquitylation target in Fzo1, lysine 398, locates to the α4 of the GTPase domain, highly conserved helix in both large and small GTPases (Daumke & Praefcke, 2016; Li et al, 2018). In the recently crystalized MGD-MFN1 structures, α4 is not part of the GTPase dimer interface, but rather sticks out at the side of each monomer (Cao et al, 2017; Yan et al, 2018), a feature also observed on our modelled MGD-Fzo1. Our results indicate that conformational changes within α4, occurring upon GTP hydrolysis, are required for Fzo1 ubiquitylation and for mitochondrial fusion. Indeed, modelling of the pre-hydrolysis state of the Fzo1 dimer revealed that α4 in wt Fzo1 shows a continuous helix. In contrast, in the post-hydrolysis state, α4 is interrupted by a short loop sequence, which harbours K398. Consistent with the fusion-impairment of the K398R mutant, the post-hydrolysis structure of its α4 does not display this interruption, suggesting inaccessibility for ubiquitin ligases. In turn, further mutating K382 in a K398R context resembled again the wt-like conformation of α4. Importantly, this rescued mitochondrial fusion by re-establishing ubiquitylation, now dependent on the

only available surrounding lysine, K387. Together, this links the conformation of α4 to the ubiquitylated state and functionality of Fzo1.

### Cdc48 and Ubp2 fail to regulate canonical ladder-like ubiquitylation in Fzo1

Concomitant with the formation of ubiquitin chains on lysine 398, α4 bending upon GTP hydrolysis also allows recruitment of Cdc48 to the Fzo1 molecules engaged at the fusion site. Indeed, despite the fact that Fzo1$^{K398R}$ is still ubiquitylated, this mutant showed impaired binding to Cdc48 and was insensitive to the hypomorphic cdc48-2 mutation, consistent with its defective fusion capacity. Importantly, this point is strengthened by the fact that Fzo1$^{K398,382R}$ (i.e., the mutant rescuing Fzo1 ubiquitylation levels) can interact again with Cdc48, whereas Fzo1$^{K398,382,387R}$ (i.e., the mutant where Fzo1 is not ubiquitylated anymore) cannot. Like Cdc48, Fzo1$^{K398R}$ is not regulated by Ubp2, another pro-fusion regulator of Fzo1. These findings support the importance of the conserved ubiquitylation pattern in recruiting cellular components necessary for fusion.

### Similarities between α4 in small and large GTPases

In small Ras GTPases, which can be anchored to membranes via C-terminal lipid anchors (Hang & Linder, 2011), it was shown that α4 plays an important role in their tight association with the membranes (Gorfe et al, 2007; Abankwa et al, 2008, 2010; Prakash et al, 2015). Specifically, positively charged residues in α4 (lysines and arginines located in positions 128 or 135 in Ras) allow electrostatic interactions with phospholipids, transiently stabilizing their orientation towards the membrane (Gorfe et al, 2007; Abankwa et al, 2008, 2010; Laude & Prior, 2008; Kapoor et al, 2012a, 2012b; Prakash & Gorfe, 2013; Mazhab-Jafari et al, 2015). Generally, this association occurs in the GTP-bound form, whereas GTP hydrolysis pushes the GTPase domain away from the membrane, perhaps allowing subsequent protein–protein interactions (Mulvaney et al, 2017; Garrido et al, 2018). Similar to the small GTPases, we identify basic residues in α4 (K382, K387, and K398) to play a role in the function of the large GTPase Fzo1. Moreover, the presence of a long α4 in the pre-hydrolysis dimer suggests it could also associate with the mitochondrial membrane in its GTP-bound form. Consistently, the higher oligomeric complex of Fzo1, which occurs after membrane tethering, is not affected by the K398R mutation. Then, bending of α4 after GTP hydrolysis might re-orient the GTPase domains of the dimers and assist in subsequent lipid mixing. The capacity to associate with the membrane and regulate fusion was also proposed for an amphipathic helix located in heptad repeat (HR) 1 of MFN1, suggesting perhaps a general feature for their functionality (Daste et al, 2018). Furthermore, Fzo1 modelled on full-length bacterial dynamin-like protein (BDLP) crystal structures in a membrane context (De Vecchis et al, 2017), as well as in an oligomeric context (Brandner et al, 2019), reveal that α4 of the Fzo1 GTPase domain is located close to the membrane (Fig S7B), consistent with the role of α4 in small GTPases.

Taken together, we propose a model in which α4 of the GTPase domain interacts with the mitochondrial membrane in its GTP-bound form and assists the fusion process. A mutation of K398 to

R probably does not interfere with membrane interaction because of its similarly positive properties. Instead, after GTP hydrolysis, K398 is required to allow Fzo1 ubiquitylation. The presence of ubiquitin chains on K398 within α4 then allows recruitment of Cdc48 to the *trans* complexes of Fzo1. Possibly, this enables to finish off the fusion process by disassembling of Fzo1 oligomers, preventing aggregation of the fusion complex (Anton et al, 2019). In fact, in SNARE-mediated membrane fusion events, the AAA-protein N-ethylmaleimide–sensitive factor (NSF), the closest relative of Cdc48, is responsible for the disassembly of *cis* SNARE complexes in the process of SNARE priming (Block et al, 1988; Malhotra et al, 1988; Puri et al, 2003). The abovementioned similarities of α4 in Ras and large GTPases, on the one hand, and of a complex disassembly role of NSF and Cdc48, on the other hand, suggest that after all similar mechanistic principles might apply to SNARE-dependent and large GTPase-dependent membrane fusion events.

# Materials and Methods

### Yeast strains and growth media

All yeast strains are isogenic to the S288c (Euroscarf). They were grown according to standard procedures to the exponential growth phase at 30°C (unless stated otherwise) on complete yeast extract–peptone (YP) or synthetic complete (SC) media supplemented with 2% (wt/vol) glucose (D), 2% (wt/vol) raffinose, or 2% galactose (wt/vol). Cycloheximide (CHX) (Sigma-Aldrich) (100 $\mu$g/ml for protein shut down from a stock of 10 mg/ml in $H_2O$) or MG132 (Calbiochem) (50 or 100 $\mu$M from a stock of 10 mM in DMSO) was added when indicated. For the analysis of Fzo1 ubiquitylation upon exclusive expression of Myc-ubiquitin, the strain YD466, isogenic to SUB328, was used (Spence et al, 1995). Fzo1 ubiquitylation dependence on SUMOylation was tested in the temperature sensitive *UBA2* mutants *uba2$^{ts}$-15* (H351P) and *uba2$^{ts}$-9* (L281S) (Schwienhorst et al, 2000). The proteasomal temperature-sensitive mutant *pre1-1* was used to test proteasome dependency (Li et al, 2011).

### Plasmids

HA-Fzo1, HA-Fzo1$^{K398R}$, HA-Fzo1$^{T221A}$, and HA-Fzo1$^{D195}$ were expressed from the centromeric plasmid pRS316 (Escobar-Henriques & Langer, 2006; Anton et al, 2013). Further HA-Fzo1 point mutants were generated by point mutagenesis in HA-Fzo1 on pRS316: K141,398,816,817R (plasmid #23), K398RT221A (#129), K141,398,816,817,79R (#200), K141,398,816,817,79,823R (#212), K141,398,816,817,79,823,484R (#213), K141,398,816,817,79,823,484,163R (#214), K141,398,816,817,79,823,484,163,263R (#215), K141,398,816,817,79,823,484,163,263,132R (#216), K382R (#382), K398,382R (#383), K398,378R (#381), K398,362R (#377), K398,371R (#379), K398,387R (#385), K398,430R (#387), K398,458R (#389), K398,459R (#391), K398,352R (#702), K398,735R (#703), K398,770R (#716), and K398,382,387R (#697). For the microscopic analysis of mitochondrial morphology, the mitochondrial matrix targeted mtGFP was expressed from the centromeric plasmid pYX142 (Westermann & Neupert, 2000). For the

amplification of gene deletion cassettes by PCR, the plasmids pFA6a-kanMX4, pFA6a-hphNT1, and pFA6a-natNT2 were used (Janke et al, 2004). Yeast mating was performed using mtGFP and mtRFP under the control of the *GAL1* promoter (Westermann & Neupert, 2000) and using 3xMyc-Fzo1 (Hermann et al, 1998) in the backbone pRS415 (Simons et al, 1987), or the corresponding point mutant 3xMyc-Fzo1$^{K398R}$, generated by point mutagenesis. To analyse the ubiquitylation linkage type in Fzo1, ubiquitin and ubiquitin harbouring the point mutation K48R were expressed from pKT10 (Tanaka et al, 1990).

### Protein steady state levels and synthesis shutoff

For analysis of protein steady state levels, total proteins from 3 OD$_{600}$ exponentially growing cells were extracted at alkaline pH (Escobar-Henriques et al, 2006) and analysed by SDS–PAGE and immunoblotting. To monitor protein turnover, cycloheximide (CHX, 100 $\mu$g/ml) was added to exponential cells. Samples of 3 OD$_{600}$ cells were collected at the indicated time points and total proteins were extracted and analysed as described above. For monitoring proteasome-dependent degradation of genomically integrated HA-Fzo1 or HA-Fzo1$^{K398R}$, additionally deleted for the multidrug transporter *PDR5* (Liu et al, 2007), the cells were treated with 50 $\mu$M MG132 for 1 h before starting a CHX chase or before alkaline lysis. Western blots were quantified using Image Quant (GE Healthcare). Proteins were detected using HA (#11867423001; Roche)-, Psd1 (gifted by S Claypool)-, Ubc6 (gifted by T Sommer)-, or Myc (#11667149001; Roche)-specific antibodies.

### Analysis of wild-type and mutant Fzo1 ubiquitylation

Fzo1 ubiquitylation was analysed as described in detail in Schuster et al (2018). In brief, 160 OD$_{600}$ cell pellets of exponentially growing cultures expressing wild-type or mutant variants of HA-Fzo1 were used to obtain crude mitochondrial extracts. After solubilization with NG310 (Lauryl Maltose Neopentyl Glycol; Anatrace), the samples were centrifuged and a portion of the supernatant was kept as input material. To immunoprecipitate HA-tagged Fzo1, the remainder of the supernatant was incubated with HA-coupled beads (Sigma-Aldrich). HA-Fzo1 was eluted in Laemmli buffer and analysed by SDS–PAGE. Proteins were transferred onto nitrocellulose membranes and subsequently immunoblotted using HA (#11867423001; Roche)-, Cdc48 (gifted by T Sommer)-, or Myc (#2276; Cell Signaling)-specific antibodies.

### Mass spectrometry

HA-Fzo1$^{K398,141,816,817R}$ was immunoprecipitated as described above with the exception that 10,000 OD$_{600}$ cells were used, precipitated with 500 $\mu$l of the HA-coupled beads (E6776; Sigma-Aldrich) and eluted with 200 mM ammonium hydroxide. The eluted protein was dried in a SpeedVac vacuum concentrator and resuspended in denaturing buffer (6 M Urea, 2 M thiourea, 20 mM Hepes, pH 8.0). The proteins were converted to peptides in a two-step protease digestion with endopeptidase LysC (Wako) and sequencing grade trypsin (Promega) (de Godoy et al, 2008). The resulting peptides were desalted and injected to a C18-reversed-phase chromatography

(75-$\mu$m column, 15 cm length in house packed, 3-$\mu$m beads Reprosil, Dr Maisch). The separated peptides were ionized on a Proxeon ion source and analysed on a Velos-Orbitrap (Thermo Fisher Scientific) mass spectrometer. The recorded spectra were analysed using the MaxQuant software package. The coverage of Fzo1 was 69.9%.

### Analysis of the interaction between HA-Fzo1 and Cdc48

Physical interactions between Cdc48 and Fzo1 or Fzo1$^{K398R}$ were analysed as previously described (Simoes et al, 2018). 160 OD$_{600}$ of yeast cells grown in complete media to the exponential growth phase were disrupted with glass beads (0.4–0.6 $\mu$m) in TBS. After centrifugation at 16,000$g$ for 10 min, the crude membrane fraction was solubilized using 0.2% NG310 for 1 h rotating at 4°C. HA-Fzo1 (or HA-Fzo1$^{K398R}$) was immunoprecipitated using Flag-coupled beads (Sigma-Aldrich) rotating over night at 4°C. Beads were washed three times with 0.2% NG310 in TBS and the precipitated protein was eluted in Laemmli buffer for 20 min shaking at 40°C. 4% of the input and 50% of the eluate fractions were analysed by SDS–PAGE and immunoblotting using HA-specific (Sigma-Aldrich) and Cdc48 (gifted by T Sommer)-specific antibodies.

### Analysis of mitochondrial morphology by epifluorescence microscopy

Yeast strains were transformed with mitochondrial-targeted GFP, grown on YP or SC media to the exponential phase and analysed as described (Escobar-Henriques et al, 2006) by epifluorescence microscopy (Axioplan 2; Carl Zeiss MicroImaging, Inc.) using a 63× oil-immersion objective. Images were acquired with a camera (AxioCam MRm; Carl Zeiss MicroImaging, Inc.) and processed with AxioVision 4.7 (Carl Zeiss MicroImaging, Inc.).

### Structural modelling of Fzo1

To analyse GTP-dependent structural changes of the GTPase domain of Fzo1, namely, its lysine 398 and $\alpha$4, Fzo1 was modelled to the crystal structures of the MGD of MFN1. Fzo1-MGD was chosen for two reasons: first its functional relation to Fzo1 in membrane fusion and second the existence of G–G dimers in the pre- and post-hydrolysis states. Structural modelling was performed using Iterative Threading ASSEmbly Refinement (i-Tasser) with standard parameters (Roy et al, 2011). Mutations of Fzo1 were modelled by amino acid exchange in the input sequence. Fzo1-MGD (amino acids 51–491; flexible linker; 826–856) was modelled on MFN1-MGD bound to GDP-BeF$_3^-$ (protein data bank identification number [PDB ID]: 5YEW) (Yan et al, 2018), to GDP-AlF$_4^-$ (PDB ID: 5GOM), or to GDP (PDB ID: 5GOE) (Cao et al, 2017).

To analyse the position of lysine 398 and $\alpha$4 of Fzo1 in a membrane context, Fzo1 was modelled on the bacterial homologue BDLP. Here, we used existing monomeric (De Vecchis et al, 2017) and oligomeric (Brandner et al, 2019) computational models of Fzo1. These were obtained from combined molecular modelling and all-atom molecular dynamics simulations in a lipid bilayer, based on the structure of GDP-BDLP (PDB ID: 2J69) (Low & Lowe, 2006).

### In vivo mating assay for assessment of fusion capacity

Analysis of mitochondrial fusion capacity was essentially performed as described (Hermann et al, 1998; Fritz et al, 2003). Exponentially growing cells of opposite mating types (BY4741 and BY4742), expressing indicated Fzo1 variants and mitochondrial matrix targeted (mt) GFP or red fluorescent protein, all under the *GAL1* promoter, were used. Cells were first grown to the exponential growth phase in SC supplemented with 2% raffinose. Expression was then induced by adding 2% galactose and incubating for 1 h and subsequently repressed by adding 2% glucose and incubation for 1 h. Then mating was induced by mixing the cells in YPD at 30°C or 37°C, as indicated. To be able to score at least 30 mating events per condition, it was necessary to wait 8 h when mating was incubated at 37°C. Fluorophore co-localization was analysed by fluorescence microscopy.

### Glycosylation assay

To test if HA-Fzo1 is glycosylated, crude mitochondrial extracts of cells expressing endogenously HA-tagged Fzo1 were prepared. Extracts were treated with Endo H$_f$ (NEB, P0703) in GlycoBuffer 3 (NEB) for 1 h at 37°C. HA-Fzo1 was eluted by adding Laemmli buffer and analysed by SDS–PAGE and immunoblotting.

### Fzo1 complex analysis by sucrose gradient centrifugation

55 OD$_{600}$ of exponentially growing yeast cells were harvested and resuspended in TBS with 1 mM PMSF and 750 mg glass beads (0.4–0.6 $\mu$m). Cells were broken upon rigorous vortexing alternated with breaks on ice for a total of 3 min. 600 $\mu$l TBS with 1 mM PMSF was added and the suspension was centrifuged for 3 min at 400$g$. The supernatant was transferred to a new centrifugation tube. The suspension was centrifuged for 10 min at 16,100$g$ and the pellet was dissolved in 50 $\mu$l (for high protein density) or 500 $\mu$l (for low protein density) of solubilization buffer (50 mM Tris–HCl, pH 7.4, 150 mM NaCl, cOmplete protease inhibitor, 1 mM PMSF) with 1% digitonin. After 1 h, the solubilized extracts were centrifuged for 10 min at 16,100$g$. 9% of the supernatant was kept as input control. Sucrose gradients were prepared in centrifugation tubes (#344059; Beckmann Coulter) as follows: 5% and 25% sucrose solutions were prepared in gradient buffer (50 mM Tris–HCl pH 7.2, 150 mM NaCl, cOmplete protease inhibitor, 0.2% Digitonin). A 5–25% gradient was generated using a gradient mixer (BioComp). The remaining 91% of the supernatant were carefully overlaid on top of the gradient. The tubes were centrifuged using an ultracentrifuge (SW41Ti rotor, 14–16 h at 4°C, 31,800 rpm). Fractions were collected in 400-$\mu$l steps from the top of the gradient and proteins were precipitated with 14.4% trichloroacetic acid on ice for 20 min. The samples were centrifuged 20 min at 16,100$g$ and 4°C, and the pellet was washed 2× with ice-cold acetone. Laemmli buffer was added to the input control and fraction pellets and samples were incubated at 50°C for 10 min and analysed by SDS–PAGE and immunoblotting, using anti-HA (#11867423001; Roche) and anti-Hsp60 (SMC-110A/B; StressMarq Biosciences Inc.) antibodies.

## DUB purification by affinity chromatography and size exclusion chromatography

USP21 was purified by affinity chromatography and size exclusion chromatography. PCR-amplified 6-His-Smt3-USP21$^{196-565}$ was cloned into pOPIN-S (Ye et al, 2011) and produced using Rosetta (DE3) pLysS (genotype: F- ompT hsdSB(rB- mB-) gal dcm (DE3) pRARE (CamR)) grown at 37°C to an $OD_{600}$ of 0.6–0.8. After cooling the culture to 18°C, gene expression was induced by 0.2 mM IPTG for 16 h at 18°C in shaking condition. The cells were harvested by centrifugation (5,000$g$, 15 min, 18°C) and then resuspended in binding buffer (20 mM Tris, pH 7.5, 300 mM NaCl, 20 mM imidazole, 2 mM $\beta$-mercaptoethanol, 1 mg/ml lysozyme, and 0.1 mg/ml DNase) and lysed by sonication. The lysates were purified by centrifugation (50,000$g$, 1 h, 4°C). The lysate was used for affinity purification using HisTrap 5 ml FF (GE Healthcare) according to the manufacturer's instructions. The 6xHis-Smt3 tag was removed by incubation with Senp1$^{415-644}$ and concurrent dialysis in binding buffer. The 6xHis-Smt3 tag and Senp1$^{415-644}$ were removed by a second round of affinity purification using HisTrap and the unbound flow-through was collected. The protein was purified by size-exclusion chromatography (HiLoad 16/600; Superdex 75 pg; buffer: 20 mM Tris, pH 7.5, 150 mM NaCl, 2 mM DTT) and the fractions were analysed on a Coomassie gel. USP21 was concentrated using VIVASPIN20 columns (Sartorius) and aliquots were frozen by liquid nitrogen and stored at −80°C.

## In vitro deubiquitylation assay

HA Immunoprecipitation, performed as described above, was used for all DUB assays (Schuster et al, 2018) using different strains, depending on the purpose: for the analysis of modification on ubiquitin itself, HA-Fzo1 of HA-Fzo1$^{K398R}$ expressed in a strain exclusively expressing Myc-tagged ubiquitin as the only ubiquitin source were used; for the removal of all ubiquitin from Fzo1, $\Delta fzo1$ cells expressing HA-Fzo1 were used; and for the analysis of DUB kinetics, cells expressing endogenously HA-tagged Fzo1 were used. For the DUB assays, instead of adding Laemmli buffer to the washed HA-coupled beads after immunoprecipitation, 35 $\mu$l DUB buffer (10× DUB buffer: 500 mM Tris pH 7.5, 500 mM NaCl, 50 mM DTT) supplemented with protease inhibitor (cOmplete EDTA-free; Roche) and 10 mM PMSF was added. Indicated amounts of purified USP21, pre-incubated in 15 $\mu$l DUB dilution buffer (25 mM Tris, pH 7.5, 10 mM DTT, and 150 mM NaCl), or 15 $\mu$l DUB dilution buffer without DUB as a control was added to the beads. The mix was incubated at 37°C for 30 min shaking. The supernatant and the beads were separated and Laemmli buffer was added to both. Eluted HA-Fzo1 from the beads was analysed by SDS–PAGE and transferred to a nitrocellulose membrane. Cleaved-off Myc-ubiquitin was analysed by tricine–SDS–PAGE and transferred to a PVDF membrane. After transfer, the PVDF membrane was denatured (6 M guanidium chloride, 20 mM Tris, pH 7.5, 1 $\mu$M PMSF, and 5 mM 2-mercaptoethanol) for 30 min, in shaking condition at room temperature and afterwards washed thoroughly before blocking and immunoblotting. HA-Fzo1 and Myc-ubiquitin were analysed using HA- or Myc-specific antibodies, respectively.

# Supplementary Information

# Acknowledgements

We would like to thank T Sommer for the Cdc48 and Ubc6 antibodies; S Claypool for the Psd1 antibody; B Westermann for the plasmids pYX142-mtGFP; pYX113-mtGFP and pYX113-mtRFP; J Shaw for pRS415-3xMyc-Fzo1; K Tanaka for the Ubiquitin plasmid and corresponding K48R mutant variant; A Buchberger for the *GAL1-CDC48* plasmid; I Woiwode for help with the purification of USP21; and A Taly for the computational models to bacterial dynamin-like protein of Fzo1. We are grateful to T Tatsuta for critical reading of the manuscript and to G Praefcke for stimulating discussions. This work was supported by grants of the Deutsche Forschungsgemeinschaft (ES338/3-1, Collaborative Research Center 1218 TP A03; to M Escobar-Henriques), the Center for Molecular Medicine Cologne (CAP14, to M Escobar-Henriques), was funded under the Institutional Strategy of the University of Cologne within the German Excellence Initiative (ZUK 81/1, to M Escobar-Henriques), and benefited from funds of the Faculty of Mathematics and Natural Sciences, attributed to M Escobar-Henriques.

## Author Contributions

R Schuster and M Escobar-Henriques wrote the manuscript, with input from all authors.
R Schuster performed most experiments and prepared the figures.
V Anton, T Simões, S Altin, performed experiments. FdB contributed to the initial observations.
T Hermanns contributed with the DUB purification.
M Hospenthal and D Komander contributed with the DUB assay.
G Dittmar performed mass spectrometry analysis.
RJ Dohmen provided intellectual input.
M Escobar-Henriques coordinated the study.

## Conflict of Interest Statement

The authors declare that the research was conducted in the absence of any commercial or financial relationships that could be construed as a potential conflict of interest.

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
