## [Reviewer comments · Life Science Alliance]

Life Science Alliance

Dual role of a GTPase conformational switch for membrane fusion by mitofusin ubiquitylation

Ramona Schuster, Vincent Anton, Tânia Simões, Selver Altin, Fabian den Brave, Thomas Hermanns, Manuela Hospenthal, David Komander, Gunnar Dittmar, R. Jürgen Dohmen, and Mafalda Escobar-Henriques

DOI: <https://doi.org/10.26508/lsa.201900476>

Corresponding author(s): Mafalda Escobar-Henriques, Institute for Genetics, Cologne Excellence Cluster on Cellular Stress Responses in Aging-Associated Diseases (CECAD), Center for Molecular Medicine Cologne (CMMC), University of Cologne

Review Timeline:	Submission Date:	2019-06-28
	Editorial Decision:	2019-08-20
	Revision Received:	2019-11-26
	Editorial Decision:	2019-12-10
	Revision Received:	2019-12-11
	Accepted:	2019-12-11

Scientific Editor: Andrea Leibfried

Transaction Report:

August 20, 2019

Re: Life Science Alliance manuscript #LSA-2019-00476

Dr. Mafalda Escobar-Henriques
Institute for Genetics, Cologne Excellence Cluster on Cellular Stress Responses in Aging-Associated Diseases (CECAD), Center for Molecular Medicine Cologne (CMMC), University of Cologne
Joseph-Stelzmann-Straße 26
Cologne 50931
Germany

Dear Dr. Escobar-Henriques,

Thank you for submitting your manuscript entitled "Dual role of a GTPase conformational switch for membrane fusion by mitofusin ubiquitylation" to Life Science Alliance. For consistency, I asked the same set of reviewers to evaluate both studies on Fzo1 you submitted to us. Their reports are appended below.

As you will see, the reviewers appreciate your analyses of the role of Fzo1 ubiquitylation for GTPase function. However, they raise some concerns that preclude publication at this stage. We would like to invite you to submit a revised version of your manuscript, addressing the reviewer concerns. Importantly, the technical issues noted need to get addressed (all reviewers) and the work needs to get better compared to the results presented in de Vecchis et al, 2017 (rev#2 and #3). Furthermore, more support for the conclusions made based on figure 3 and 5 are needed (rev#1). Point 4 of rev#3 does not need to get mandatorily addressed for publication here.

Thank you for this interesting contribution to Life Science Alliance. We are looking forward to receiving your revised manuscript.

Sincerely,

B. MANUSCRIPT ORGANIZATION AND FORMATTING:

Reviewer #1 (Comments to the Authors (Required)):

The mechanism of mitochondrial outer membrane fusion and its regulation are poorly understood. In yeast, a single dynamin-related protein, Fzo1, is required for outer membrane fusion along with Ugo1. It has been previously reported that Fzo1 ubiquitination serves a complex regulatory role in Fzo1-mediated fusion, with several unique players implicated. This group has reported that Fzo1-K398 is one of the lysine residues subject to ubiquitination; however, the role of this post-translational modification (PTM) in Fzo1-activity was unclear, as substitution of R at this position did not inactivate Fzo1, though it did alter the ubiquitination pattern on SDS-PAGE. In this manuscript, further characterization of Fzo1-K398R suggests that loss of ubiquitin at this position results in a decreased interaction with Cdc48 and addition of the PTM at an adjacent lysine residue, which signals for rapid proteasomal degradation of Fzo1-K398R.

The manuscript is a little difficult to follow and some suggestions are outlined below aimed to improve readability. Ultimately, the defects associated with loss of ubiquitination at this site are minor, as all mutants are only defective at non-permissive temperatures. Nonetheless, these types of mutant alleles can be very useful in functional analysis. This work demonstrates that helix four is a target for ubiquitination and that in the absence of a preferred site, the E3 ubiquitin ligase will attach ubiquitin to a secondary or even tertiary site. Fusion activity is restored when ubiquitin is located on K387 but not K382, which may suggest that the ubiquitin needs to be near the C-terminal end of helix four. In turn, the placement of the ubiquitin may be important for Cdc48 binding to Fzo1. Unfortunately, this is not tested in the current version of this manuscript.

Figure 1: This group previously demonstrated that Fzo1-K398R partially rescued mitochondrial fusion activity in cells lacking *fzo1*. In this manuscript, mitochondrial morphology was scored in $\Delta fzo1$ cells expressing Fzo1-K398R and a defect in fusion activity was observed at higher temperatures. This was confirmed with the yeast mating assay for mitochondrial fusion. The mating assay further revealed that the defect could not be complemented by the presence of a wild type Fzo1 in trans. Authors conclude that fusion is impaired under cellular stress conditions.

1a. Is this a general stress response, or is it specific to proteostasis?

Figure 2: Levels of Fzo1 were assessed following protein synthesis arrest by cycloheximide in the presence and absence of the proteasomal inhibitor, MG132. Results are consistent with the conclusion that the Fzo1-K398R ubiquitination pattern is associated with rapid proteasomal-dependent turnover of Fzo1-K398R.

Figure 3: Normally, Fzo1 is protected from proteasomal turnover in two ways: removal of ubiquitin by Ubp2 and association with Cdc48. Wild type Fzo1 is turned over more rapidly in the absence of Ubp2, but this is not the case for Fzo1-K398R. This suggests that Ubp2 does not recognize the degradation-promoting ubiquitin modifications.

3a. Given the low levels of Fzo1-K398R, it might be useful to perform this experiment in cells also expressing Fzo1-K398R from the CEN-ARS plasmid.

Fzo1-K398R co-immunoprecipitated less Cdc48 compared to wild type Fzo1. In cells lacking functional Cdc48, there was no change in Fzo1-K398R levels, whereas wild type Fzo1 decreases. These data are consistent with the conclusion that Fzo1-K398R is more rapidly turned over due to a decreased interaction with Cdc48

3b. There is still a small amount of Cdc48 that co-immunoprecipitated with Fzo1-K398R,

suggesting a reduced affinity. Does overexpression of Cdc48 restore protein stability or fusion activity in cells expressing Fzo1-K398R? Alternatively, can fusion activity be rescued by tethering Cdc48 to the terminus of Fzo1 or using the Ubp7 fusion that was reported in Anton 2013?

Figure 4: Authors also search for an explanation for the unusually Fzo1 banding pattern observed in wild type cells. Results indicate that all higher migrating species are eliminated upon treatment with a deubiquitinating enzyme and that the removed ubiquitin species are not modified. In addition, the pattern is examined in cells lacking several ubiquitin modifiers and no change is observed, indicating that the bands do not represent an association between Fzo1 and these proteins.

Authors suggest that the unique migration of these species is due to SDS-resistant conformational differences in Fzo1. As evidence, two different SDS-PAGE conditions are presented where the apparent molecular weight is different.

4a. The molecular weight for each species should be estimated based on migration of the molecular weight standards under each condition to emphasize this point.

4b. Authors might consider moving this to supplement and discussion of the data to the beginning of the paper. Perhaps with Figure 1A where the banding pattern is reported/confirmed. For this reviewer, it does not fit in the middle of the story.

Figure 5: Given that Fzo1-K398R is rapidly turned over by the proteasome, ubiquitin must be attached to another lysine residue. Variants of Fzo1 with substitution of arginine at one of eight lysine residues near K398R were examined for increased steady state levels of Fzo1. Only one resulted in more Fzo1, as expected for a site important for ubiquitin modification (K382). A version of Fzo1 with both lysine residues changed to arginine (Fzo1-K398R;K382R) had a ubiquitination migration pattern similar to wild type and some mitochondrial fusion activity in cells lacking fzo1.

5a. The previous experiments argue that Fzo1-K398R is turned over more rapidly due to a defective interaction with Cdc48. This leads to the hypothesis that the double substitution (Fzo1-K398R;K382R) may have restored binding to Cdc48. This should be tested to support the conclusions.

In Fzo1-K398R;K382R, another lysine residue must be the target of ubiquitination. The role of K387 was tested by creating a triple K-R Fzo1 protein (Fzo1-K398R;K382R;K387R), which was shown to have an aberrant ubiquitination pattern and no mitochondrial fusion activity at 37 degrees.

5b. Does Fzo1-K398R;K382R;K387R show fusion activity at permissive temperature?

5c. Does Fzo1-K398R;K382R;K387R interact with Cdc48?

Figure 6: Molecular modeling predicts that the double mutant may restore the conformation of helix four in the GTPase domain to include an interruption/loop that may be lost in Fzo1-K398R. Authors connect this to the predicted structures of Fzo1 based on BDLP and Mfn1-MGD where helix 4 is predicted to undergo a similar conformational change. Authors hypothesize that this indicates that helix four is unlikely to be subject to ubiquitination before GTP hydrolysis. This reviewer does not follow the logic behind this conclusion. To support this, the interaction of Fzo1 with Cdc48 in different nucleotide states would need to be tested, which is a very challenging set of experiments.

Reviewer #2 (Comments to the Authors (Required)):

Overall impression:

Schuster et al. utilize a combination of techniques to meticulously examine the contributions of K398 ubiquitination during mitochondrial fusion. For the most part the data are very clean, well presented and correctly controlled. In the end, they find that the K398R mutation that blocks fusion likely does so by altering the conformation of the $\alpha 4$ helix and prevents it from adopting the bent conformation seen after GTP hydrolysis. This idea is strongly backed by the double mutant data with K398R, K283R, which restores the WT-like kink in $\alpha 4$ and restores fusion. I recommend this manuscript be accepted for publication with minor revisions.

Summary:

The authors set out to determine the molecular mechanism for the required Fzo1 ubiquitination during mitochondrial fusion. A main site of ubiquitination for fusion was previously identified as lysine 398. When mutated (K398R), the unique banding pattern of Fzo1 ubiquitination is lost and instead has a canonical ubiquitin ladder, which leads to its rapid degradation. When the authors paired the K398R mutation with GTPase mutations impairing GTP binding and GTP hydrolysis, they see a loss of ubiquitination on Fzo1, raising the question; can Fzo1 K398R still fuse mitochondria, even at reduced protein levels? To this end, they nicely show that under heat stress the K398R mutant has highly fragmented mitochondria. The K398R mutant was also shown to be able to form a trans complex of oligomers, suggesting a downstream step of fusion is being disrupted. Using an elegant *in vivo* fusion assay, the authors then argue the K398R mutant can disrupt fusion with WT Fzo1 under heat stress, showing proper ubiquitination is required on both dimers of the trans complex. Next the authors wanted to determine if the K398R fusion defect was due to a change in ubiquitination status or due to lower steady state protein levels compared to WT Fzo1. To overcome the lower protein levels from proteasome-dependent turnover they overexpressed the Fzo1 K398R on a plasmid to obtain steady state levels comparable to WT and there is still a fusion defect at 37C as assessed by the % tubular mitochondria. The results were also reproduced by inhibiting the proteasome. The K398R mutant clearly has a fusion defect not attributable to the lower levels of the protein.

The authors then show that K398R is not modulated by downstream partners in the fusion process. The DUB Ubp2 normally removes degradative ubiquitin chains from WT Fzo1, but the authors show that Ubp2 deletion does not stabilize K398R. They then examine the interaction with the AAA ATPase Cdc48, a pro-fusion factor that interacts with Fzo1 and in part prevents proteasomal turnover of the protein. They see a reduced interaction with CDC48 compared to WT and more or less no change in steady state levels when Cdc48 is rendered inactive, suggesting K398R does not properly associate with Cdc48 for fusion to proceed correctly.

After establishing that the 3 HMW bands of WT Fzo1 are in fact ubiquitinated by *in vitro* cleavage with Usp21, the authors also tested for Ub-like modifications, glycosylation and modification of Ub itself, and none of these are present in the modified Fzo1. Instead they account for the non-canonical band shifts of ubiquitinated Fzo1 as forms resistant to complete denaturation during SDS-PAGE resolution.

With the structure of the human Fzo1 homolog mitofusion-1 minimal GTPase domain known, the authors were able to model in Fzo1 and see where K398R resides. Given K398R is still ubiquitinated they examined additional lysine mutations at all of the proximal 8 lysine residues, 3 highly conserved lysines and all the lysines identified by mass spec to be ubiquitinated. When K398R was paired with the K382R mutant (and only this mutant) the steady state levels of the protein increased to about half of WT, exhibited the WT ubiquitination pattern and most importantly rescued the K398R fusion defect. Based on their structural modeling they attribute this to the $\alpha 4$ conformational changes where both of these lysines reside. Their data strongly support the model that the K398R mutation alone makes $\alpha 4$ rigid mimicking the GTP-bound state and likely prevent/greatly impedes the transition to the bent $\alpha 4$ helix observed after GTP hydrolysis, which is necessary for correct ubiquitination and recruitment of Cdc48. In the discussion they make a nice observation of the parallels between this system and the disassembly of SNARE complexes.

Issues to resolve:

- Scale bar in Figure 1B
- Figure 1C: include a trans oligomer mutant here as reference along with a representative experiment of the primary data in the supplemental figure.
- The methodology of the experiment in Figure 1D seems flawed; the fluorescent reporter proteins and the Fzo1 variants are all expressed from galactose promoters and the fusion assessment is carried out 9 h after turning off expression by switching to glucose. The switch from galactose to glucose rapidly and tightly represses GAL1-promoter based expression and with Fzo1 K398R being degraded by the proteasome with ~30 min half-life (Figure 2A and Anton et al. 2013); it is hard to imagine that any mutant Fzo1 is still present even by the time yeast form zygotes. One control that would resolve this is to carry out the assay as usual but grow individual haploid strains in YPD instead of mixing them and then assess the Fzo1 proteins levels after 8 h of growth. In the end, I'm not sure if this experiment is even necessary for the story.
- The cycloheximide data (Figure 2A) showing the mutant form is rapidly degraded compared to the WT is explained as a novel result even though the same group previously published a K398R chase in Anton et al. 2013. Please adjust the results and discussion text accordingly.
- Figure 3E. Can a long exposure of this be shown? Even better a CHX chase.
- "Moreover, the two heavier ubiquitylated bands correspond to K48-linked chains conjugated to lysine 398 in Fzo1." This is a statement made early in the results section, yet the supplemental figure 2A shows these bands persist in the K48R plasmid expressing strain. Is an alternative chain type formed on WT Fzo1 in the K48R mutant or is the pattern from the endogenous WT ubiquitin? Is fusion carried out normally in the K48R expressing strain?
- Why does the amount of modified Fzo1 change in the uba2 ts mutants?
- How does the model of $\alpha 4$ interacting with the membrane compare with the one proposed by De Vecchis et al. 2017?

Reviewer #3 (Comments to the Authors (Required)):

This manuscript reports efforts to understand the mechanism by which ubiquitylation regulates the yeast mitofusin Fzo1p during mitochondrial fusion. Mitofusins generally appear to show a ubiquitylation pattern that is conserved between yeast and mammals though the relevance of this pattern to fusion vs degradation is unclear (and sequence variations are high). The authors use changes in the Ub pattern as a surrogate for active fusion forms and focus around the known Ub residue Lysine 398. They demonstrate that this mutant is temperature sensitive for fusion and while Ub still occurs, the characteristic Fzo1 pattern on SDS-PAGE differs markedly. This pattern is

restored by a suppressor mutation in K387R. They rationalise this effect through structural modelling of Fzo1 by suggesting that the alpha-helix 4 undergoes a conformational switch that is important for recognition by the deubiquitylase Ubp2 and Cdc48/p97 for disassembly. The manuscript is well-constructed manuscript and the overall conclusions make sense based on the experiments conducted. The model for membrane fusion is useful for future tests and to determine how well this is conserved in evolution. My only, but significant, concern is that the overall mechanistic conclusion is based on structural modelling approaches that may be incorrect.

1. More clarity is required into the structural modelling and the changes in alpha4 as a result of conserved aa changes. Can the authors provide a clear structural explanation for this and more details into the modelling approach used?

2. How does the authors' model compare with that presented by De Vecchis and colleagues? *Sci rep* 2017 <https://www.nature.com/articles/s41598-017-10687-2>. In that work, it was shown that a distant mutation affected the Ub pattern but this could be rescued by making a salt bridge reversal mutant of residues. This indicates that sequence changes can result in large conformational shifts that may be misinterpreted.

3. The sequence of Fzo1 and mammalian Mfn1 is not well conserved so it is difficult to surmise if the structures and functionality can be modelled in such detail. Can the authors clarify as to whether the residues that are Ub'd here are present in the mammalian sequence? If not, how do they interpret the results?

4. The Ub pattern of Fzo1 is investigated in Fig. 4 and the authors suggest that this is a result of higher Fzo1 conformers that are resistant to SDS-PAGE. While not critical, the authors could support this in more detail by undertaking a urea-SDS-PAGE approach to see if the forms dissociate.

Reply to Reviewer #1:

We would like to thank the reviewer for acknowledging the usefulness of mutant alleles for functional analyses. We are also grateful for the careful analysis of our manuscript, as well as the many valuable suggestions. We fully addressed this reviewer's advice regarding the readability as well as the experimental concerns.

1. **“Figure 1”**: *“This group previously demonstrated that Fzo1-K398R partially rescued mitochondrial fusion activity in cells lacking fzo1. In this manuscript, mitochondrial morphology was scored in delta fzo1 cells expressing Fzo1-K398R and a defect in fusion activity was observed at higher temperatures. This was confirmed with the yeast mating assay for mitochondrial fusion. The mating assay further revealed that the defect could not be complemented by the presence of a wild type Fzo1 in trans. Authors conclude that fusion is impaired under cellular stress conditions.*

1a. Is this a general stress response, or is it specific to proteostasis? “

Indeed, thanks to this suggestion, we can now show that K398R mutation impairs Fzo1 function upon general stress stimuli (Fig. 1C). In addition to heat stress, we have tested the effect of several other cellular stress conditions (*i.e.* salt stress (0.5 M NaCl), sub-lethal doses of cycloheximide (0.5 µg/ml) and mitochondrial membrane potential impairment (10 µM CCCP)) on mitochondrial tubulation. Consistent with high temperature, cells expressing HA-Fzo1^{K398R} revealed strong tubulation defects upon these stresses, while in cells expressing wild-type HA-Fzo1 mitochondrial morphology was not affected.

2. **“Figure 3”**: *“Normally, Fzo1 is protected from proteosomal turnover in two ways: removal of ubiquitin by Ubp2 and association with Cdc48. Wild type Fzo1 is turned over more rapidly in the absence of Ubp2, but this is not the case for Fzo1-K398R. This suggests that Ubp2 does not recognize the degradation-promoting ubiquitin modifications.*

3a. Given the low levels of Fzo1-K398R, it might be useful to perform this experiment in cells also expressing Fzo1-K398R from the CEN-ARS plasmid.”

As suggested, we have now performed a cycloheximide chase using cells with adjusted HA-Fzo1^{K398R} levels, by ectopic expression, which supported our conclusions (new Fig S5A).

3. **“Figure 3”**: *“Fzo1-K398R co-immunoprecipitated less Cdc48 compared to wild type Fzo1. In cells lacking functional Cdc48, there was no change in Fzo1-K398R levels, whereas wild type Fzo1 decreases. These data are consistent with the conclusion that*

Fzo1-K398R is more rapidly turned over due to a decreased interaction with Cdc48

3b. There is still a small amount of Cdc48 that co-immunoprecipitated with *Fzo1-K398R*, suggesting a reduced affinity. Does overexpression of Cdc48 restore protein stability or fusion activity in cells expressing *Fzo1-K398R*? Alternatively, can fusion activity be rescued by tethering Cdc48 to the terminus of *Fzo1* or using the *Ubp7* fusion that was reported in Anton 2013? "

To address this point, we have tried both approaches suggested. Please refer to the attached figures.

First, we overexpressed Cdc48 using the *GAL1* inducible promoter. However, at 37 °C, this already shifted the mitochondrial network of wild-type cells towards fragmentation. Thus, this approach did not allow to test the reviewer's suggestion. It should be noted that Cdc48 is already one of the most abundant proteins in the cell, with multiple functions, perhaps not tolerating any overexpression.

Fig. R1 - Effect of Cdc48 overexpression in mitochondrial morphology. Mitochondrial morphology (**A**) and Cdc48 expression (**B**) of cells expressing Cdc48 under the control of the *GAL1* promoter, and either HA-Fzo1 or HA-Fzo1^{K398R} integrated into the genome, together with Su9 mCherry (Anton 2019, LSA). Cells were grown in selective media containing 2% glucose (D), 2% raffinose (R) or 2% raffinose supplemented with 2% galactose for one hour (G), and either analysed for morphology (**A**) or used to prepare total cellular extracts and were analysed by SDS-PAGE and Western blot, using a Cdc48-specific antibody (**B**).

Second, we fused Cdc48 to the C-terminus of HA-Fzo1 expressed from a centromeric vector. However, while the expression of HA-Fzo1 rescued the mitochondrial morphology of $\Delta fzo1$ cells, expression of HA-Fzo1-Cdc48 did not (A,B). As we observed a very low expression level of this construct, we then cloned Fzo1-Cdc48 into a 2 μ vector. Once again, despite a rescue of the expression level, also this

construct did not rescue the mitochondrial morphology (C,D). Therefore, this approach was also not suitable to answer the reviewer's suggestion.

Fig. R2 - Effect of Fzo1-Cdc48 fusion in mitochondrial morphology. (A+B) Expression of Fzo1-Cdc48 from a centromeric plasmid. Cells lacking endogenous Fzo1, transformed with HA-Fzo1 (closed arrowhead) or HA-Fzo1-Cdc48 (open arrowhead) from the centromeric pRS316 plasmid, also expressing Su9 mCherry, were analysed for mitochondrial morphology (A) and were also analysed for protein levels, by SDS-PAGE and Western blot, using an HA-specific antibody (B). (C+D) Expression of Fzo1-Cdc48 from the 2 μ YEplac195 plasmid, analysed as in A and B.

4. “Figure S2” (old Figure 4): “Authors also search for an explanation for the unusually Fzo1 banding pattern observed in wild type cells. Results indicate that all higher migrating species are eliminated upon treatment with a deubiquitinating enzyme and that the removed ubiquitin species are not modified. In addition, the pattern is examined in cells lacking several ubiquitin modifiers and no change is observed, indicating that the bands do not represent an association between Fzo1 and these proteins.

Authors suggest that the unique migration of these species is due to SDS-resistant conformational differences in Fzo1. As evidence, two different SDS-PAGE conditions are presented where the apparent molecular weight is different.

4a. The molecular weight for each species should be estimated based on migration of the molecular weight standards under each condition to emphasize this point.”

The apparent molecular weight of the ubiquitylated forms of Fzo1 upon different SDS-PAGE conditions has now been added to the second sub-chapter of the results section: “No other UBL accounts for the large molecular shift in Fzo1 ubiquitylation”

5. “Figure S2” (old Figure 4): *“4b. Authors might consider moving this to supplement and discussion of the data to the beginning of the paper. Perhaps with Figure 1A where the banding pattern is reported/confirmed. For this reviewer, it does not fit in the middle of the story.”*

As suggested, we have now moved it to the supplement of Figure 1, as the new Fig S2.

6. “Figure 4” (old Figure 5): *Given that Fzo1-K398R is rapidly turned over by the proteasome, ubiquitin must be attached to another lysine residue. Variants of Fzo1 with substitution of arginine at one of eight lysine residues near K398R were examined for increased steady state levels of Fzo1. Only one resulted in more Fzo1, as expected for a site important for ubiquitin modification (K382). A version of Fzo1 with both lysine residues changed to arginine (Fzo1-K398R;K382R) had a ubiquitination migration pattern similar to wild type and some mitochondrial fusion activity in cells lacking fzo1.*

5a. The previous experiments argue that Fzo1-K398R is turned over more rapidly due to a defective interaction with Cdc48. This leads to the hypothesis that the double substitution (Fzo1-K398R;K382R) may have restored binding to Cdc48. This should be tested to support the conclusions.”

Indeed, we tested HA-Fzo1^{K398R,K382R} interaction with Cdc48 by co-immunoprecipitation and show that this Fzo1 mutant has restored binding capacity to Cdc48 (Fig 4B), as expected. This supports the idea that reduced Cdc48 binding increases the turn-over of Fzo1^{K398R}, which we now clearly discuss in the sixth sub-chapter of the results section: “Conformational readjustment in $\alpha 4$ of Fzo1 can compensate for the ubiquitylation defect in Fzo1^{K398R}”.

7. “Figure 4” (old Figure 5): *“5b. Does Fzo1-K398R;K382R;K387R show fusion activity at permissive temperature? “*

We show now that Fzo1^{K398R,K382R,K387R} cells show fusion activity at permissive temperature (Fig S7A). For clarity, the morphology analyses of the triple mutant at non-permissive temperature has been repeated with controls and moved from old Fig

5D to new Fig 5D. Also, for the sake of completeness, we provide the morphology analysis at permissive temperature for the double mutant Fzo1^{K398R,K382R} in Fig S6E.

8. “Figure 5” (old Figure 6): *“5c. Does Fzo1-K398R;K382R;K387R interact with Cdc48?”*

We tested HA-Fzo1^{K398R,K382R,K387} interaction with Cdc48 by co-immunoprecipitation and clearly show impaired binding of Cdc48, as expected (Fig 5C).

Together, the coIP results from Fig. 4B and Fig. 5C support the conclusion that ubiquitylation on $\alpha 4$ is important for Cdc48 binding to Fzo1, as stated in the last sub-chapter of the results section: “Off-site rescue of Fzo1 ubiquitylation is required for functionality”.

9. “Figure 5” (old Figure 6): *“Molecular modeling predicts that the double mutant may restore the conformation of helix four in the GTPase domain to include an interruption/loop that may be lost in Fzo1-K398R. Authors connect this to the predicted structures of Fzo1 based on BDLP and Mfn1-MGD where helix 4 is predicted to undergo a similar conformational change. Authors hypothesize that this indicates that helix four is unlikely to be subject to ubiquitination before GTP hydrolysis. This reviewer does not follow the logic behind this conclusion. To support this, the interaction of Fzo1 with Cdc48 in different nucleotide states would need to be tested, which is a very challenging set of experiments.”*

We realize that this description was confusing and we would like to apologize for this. The corresponding text (*i.e.* the last two chapters of the results) has been remodelled. We now clearly state previous results from Anton et al., 2011, showing that Fzo1 ubiquitylation occurs after GTP hydrolysis.

Reply to Reviewer #2:

We are thankful to Reviewer #2 for the support to publish and for such a clear statement of our main results. We appreciate the suggestions made by this reviewer, which we have addressed carefully.

1. *“Scale bar in Figure 1B”*
A scale bar was now added to Fig 1B.

2. *“Figure 1D (old figure 1C): include a trans oligomer mutant here as reference along with a representative experiment of the primary data in the supplemental figure.”*

Higher oligomerization of Fzo1 only occurs after mitochondrial concentration, allowing interactions *in trans* (Anton et al., 2011). Thus, we now included in Fig S3A a control in low density of mitochondria where *trans* oligomerization does not occur. Additionally, we added a representative experiment of Fig 1D in the supplemental figure (Fig S3B).

3. *“The methodology of the experiment in Figure 1D seems flawed; the fluorescent reporter proteins and the Fzo1 variants are all expressed from galactose promoters and the fusion assessment is carried out 9 h after turning off expression by switching to glucose. The switch from galactose to glucose rapidly and tightly represses GAL1-promoter based expression and with Fzo1 K398R being degraded by the proteasome with ~30 min half-life (Figure 2A and Anton et al. 2013); it is hard to imagine that any mutant Fzo1 is still present even by the time yeast form zygotes. One control that would resolve this is to carry out the assay as usual but grow individual haploid strains in YPD instead of mixing them and then assess the Fzo1 proteins levels after 8 h of growth. In the end, I'm not sure if this experiment is even necessary for the story.”*

We realize that this control is essential and have now included it as the new Fig S3C. We show that both Fzo1 and Fzo1^{K398R} are still present 8 hours after synthesis shut-off by switching to glucose. Importantly, we perform this experiment after repression of promoter activity by addition of glucose to prevent artefacts and guarantee that we only monitor fusion by Fzo1 proteins that are already present before mating and not newly synthesized. Moreover, it should be noted that at 37 °C mating is very inefficient, thus, we needed to wait 8 hours in order to be able to score at least 30 mating events per condition analysed. This is now clearly stated in the material and methods.

4. *“The cycloheximide data (Figure 2A) showing the mutant form is rapidly degraded compared to the WT is explained as a novel result even though the same group previously published a K398R chase in Anton et al. 2013. Please adjust the results and discussion text accordingly.”*

We realize that our wording was easy to misunderstand. We have previously shown that Fzo1^{K398R} is rapidly degraded when compared with wild-type Fzo1 (Anton et al., 2013). What we now newly show is that this degradation is dependent on the proteasome. We apologize for this mistake and we have carefully corrected it in the fourth sub-chapter of the results section: “Canonical ladder-like ubiquitylation destabilizes Fzo1 and possesses no fusogenic activity”.

5. *“Figure 3E. Can a long exposure of this be shown? Even better a CHX chase.”*

A longer exposure of the signal has been added to Fig 3E. Additionally, we have now included a cycloheximide chase of the same strains, confirming the insensitivity of Fzo1^{K398R} to Cdc48 (Fig S5B).

6. *“Moreover, the two heavier ubiquitylated bands correspond to K48-linked chains conjugated to lysine 398 in Fzo1.” This is a statement made early in the results section, yet the supplemental figure 2A shows these bands persist in the K48R plasmid expressing strain. Is an alternative chain type formed on WT Fzo1 in the K48R mutant or is the pattern from the endogenous WT ubiquitin? Is fusion carried out normally in the K48R expressing strain?”*

Indeed, the strains that were used to analyse the K48-linked chains in Fzo1 still express wild-type ubiquitin, as now clearly stated in the fourth sub-chapter of the results section: “Canonical ladder-like ubiquitylation destabilizes Fzo1 and possesses no fusogenic activity”. Furthermore, we tested mitochondrial morphology (new Fig S4B), which showed that the expression of K48R mutated ubiquitin generally decreased mitochondrial tubulation, as expected.

Additionally, we analysed Fzo1 ubiquitylation and mitochondrial morphology upon overexpressing K48R ubiquitin using the *CUPI* promoter, as shown below. This resulted in higher impairment of ubiquitin chains and fragmentation of mitochondria, as expected. However, we would prefer not including it in the main text, because K48 is essential and could thus cause secondary effects indirectly affecting mitochondrial dynamics.

Fig. R3 - Analysis of the ubiquitin linkage-type of Fzo1 and Fzo1^{K398R} upon overexpression of Myc-ubiquitin and its mutant variant K48R using a copper promoter. (A) Mitochondrial morphology (upper panel) and whole cell extracts (bottom panel) in cells expressing endogenously HA-tagged Fzo1 (wt) or Fzo1^{K398R} (K398R) and ectopically expressing Myc-ubiquitin wt (Ub wt) or with a K48R mutation (Ub K48R) under control of the *CUP1* promoter. Cells as indicated were grown at 30 °C in presence or absence of copper, as indicated, and analysed for mitochondrial

morphology as described in Fig 1B. and for ubiquitin expression by SDS-PAGE and Western blot, using Ub-, Myc- and HA-specific antibodies.

(B) Crude mitochondrial extracts from cells as indicated in A were solubilized and analysed by SDS-PAGE and Western blot, using HA-, Myc- and Ub-specific antibodies. **(C, D)** Preliminary experiment to test for copper inducible overexpression of Myc-ubiquitin. Crude mitochondrial extracts (C) and whole cell extracts (D) from cells as in A, grown in presence or absence of copper, as indicated. Over-expression of the two ubiquitin variants was induced by supplying the media with 50 μ M of CuSO₄, for 4h.

7. “Why does the amount of modified Fzo1 change in the *uba2* ts mutants?”

Indeed, we observed that the deletion or point mutation of the other post-translational modifiers, including *uba2^{ts}*, generally increased the ubiquitylation of Fzo1. This might depend on the general balance of PTM reactions, which is now stated in the second sub-chapter of the results section: “No other UBL accounts for the large molecular shift in Fzo1 ubiquitylation”.

8. “How does the model of $\alpha 4$ interacting with the membrane compare with the one proposed by De Vecchis et al. 2017?”

Indeed, the models from A. Taly, presented both in De Vecchis and in Brandner 2019, constitutes an excellent tool to analyse Fzo1 in a membrane context. These computational models are based on a full-length structure of the bacterial homologue BDLP bound to GDP and were obtained by combined molecular modelling and all-atom molecular dynamics simulations in a lipid bilayer. Thus, thanks to the suggestions by the reviewers, we have now taken the BDLP-based model into account. As we show in Fig S7B, the location of helix 4 in this model is in close proximity to the membrane, supporting the idea of a similar function of helix 4 in small and large GTPases.

However, the corresponding lipid bilayer simulation analysis of pre-hydrolysis state is not available. In turn, dimeric structures of the pre- and post-hydrolysis transition states exist for the truncated version of MFN1, the functional homologue of Fzo1. Given that we focused on the GTPase domain, these structures were chosen to analyse GTP-dependent structural changes of the GTPase domain.

Reply to Reviewer #3:

We are thankful that this reviewer found our manuscript well-constructed with overall conclusions that make sense. Importantly, the model by De Vecchis and colleagues also place

helix 4 and K398 in proximity to the membrane, emphasizing its importance and supporting the idea that helix 4 of Fzo1 has a similar role as in small GTPases. As mentioned by the reviewer, we propose a model for membrane fusion, useful for future tests. In particular, we agree with the reviewer that in mammals the proposed mechanism and the relevance of the conserved ubiquitylation pattern for fusion versus degradation still needs to be investigated.

1. *“More clarity is required into the structural modelling and the changes in alpha4 as a result of conserved aa changes. Can the authors provide a clear structural explanation for this and more details into the modelling approach used?”*

We apologize for the lack of clarity which we now corrected in the material and methods.

Structural modelling was performed using Iterative Threading ASSEmblY Refinement (i-Tasser) with standard parameters. Mutations of Fzo1 were modelled by amino acid exchange in the input sequence. Importantly, the predictions based on the structural models were observed experimentally. In our view this justifies the model we propose here.

2. *“How does the authors' model compare with that presented by De Vecchis and colleagues? Sci rep 2017 <https://www.nature.com/articles/s41598-017-10687-2>.”*

As mentioned in point 8 of reviewer 2, as we were interested in dynamic changes in the GTPase domain, we modelled Fzo1 on the functionally related MFN1-MGD, instead of using the BDLP-based structures by De Vecchis and colleagues. Nevertheless, a comparison of our model with the one from De Vecchis and colleagues shows that our hypotheses are in agreement with their structure. In the new Fig S7B we used the structures kindly provided by A. Taly to locate helix 4 of the GTPase domain, which we found in close proximity to the membrane, supporting our model.

In that work, it was shown that a distant mutation affected the Ub pattern but this could be rescued by making a salt bridge reversal mutant of residues. This indicates that sequence changes can result in large conformational shifts that may be misinterpreted.”

The possibility to rescue Fzo1 ubiquitylation with distal mutations is explained by the fact that Fzo1 ubiquitylation only occurs at a late stage of the fusion process. Therefore, it is not surprising that many mutations preventing previous steps in fusion will abolish Fzo1 ubiquitylation. In fact, the mechanistic features of the salt-bridge rescue are now further explained in our manuscript recently accepted in LSA (Anton et al., 2019).

In the present manuscript, however, we directly test the structural predictions on the ubiquitylated residues. Therefore, at least for Fzo1, we are confident that we are analysing a direct effect.

3. *“The sequence of Fzo1 and mammalian Mfn1 is not well conserved so it is difficult to surmise if the structures and functionality can be modelled in such detail. Can the authors clarify as to whether the residues that are Ub'd here are present in the mammalian sequence? If not, how do they interpret the results?”*

The structure of MFN1 has a lysine residue (K269) at the same position as lysine 398 in Fzo1. A GDP-bound-structure of MFN2 is now available, revealing an arginine residue (R294) at the same position as lysine 398 in Fzo1. Albeit not offering the possibility for ubiquitylation this positively charged residue might be involved in membrane interactions as observed in small Ras GTPases. Importantly, the interruption of helix 4 into 4a and 4b is visible in all GDP-bound structures.

Fig. R4 – Comparison of Fzo1 K398 to analogue residues in MFN1 (K269) and MFN2 (R294). Positively charged residues at the C-terminus of helix 4 in Fzo1-MGD modelled on MFN1-MGD bound to GDP (5GOE), MFN1-MGD bound to GDP (5GOE) and MFN2 bound to GDP (6JFK). K398 in Fzo1 and analogue residues in MFN1 (K269) and MFN2 (R294) are shown as yellow sticks, helix 4 is shown in red.

Moreover, a comparison between the helices 4 of Fzo1 modelled on MFN1-MGD and of MFN1-MGD itself has been performed. As presented in the figure below, alpha 4 appears to undergo similar changes in MFN1 and in Fzo1. However, the role of ubiquitylation and the function of the conformational changes of this helix in MFN1 remain highly speculative. Therefore, we would prefer not to include the two figures R3 and R4 in the manuscript.

Fig. R5 – Analysis of GTP-hydrolysis effects in helix 4 of MFN1 and Fzo1. Zoom-ins of helix 4, in red, of Fzo1-MGD and MFN1-MGD, bound to GDP-BeF₃⁻ or GDP-AIF₄⁻.

4. *“The Ub pattern of Fzo1 is investigated in Fig. 4 and the authors suggest that this is a result of higher Fzo1 conformers that are resistant to SDS-PAGE. While not critical, the authors could support this in more detail by undertaking a urea-SDS-PAGE approach to see if the forms dissociate.”*

Given that this point was not critical, we accepted the suggestion by the editor not to analyse it.

December 10, 2019

RE: Life Science Alliance Manuscript #LSA-2019-00476R

Dr. Mafalda Escobar-Henriques
Institute for Genetics, Cologne Excellence Cluster on Cellular Stress Responses in Aging-Associated Diseases (CECAD), Center for Molecular Medicine Cologne (CMMC), University of Cologne
Joseph-Stelzmann-Straße 26
Cologne 50931
Germany

Dear Dr. Escobar-Henriques,

Thank you for submitting your revised manuscript entitled "Dual role of a GTPase conformational switch for membrane fusion by mitofusin ubiquitylation". As you will see, the reviewers appreciate the changes introduced in revision, and we would thus be happy to publish your paper in Life Science Alliance pending final revisions necessary to meet our formatting guidelines.

- please add callouts in the manuscript text to Fig S4B and S7B
- there is callout for S4F, but there is no panel F in legend or in the figure; please fix
- please add the mass-spec results that are discussed but not shown

A. FINAL FILES:

-- Summary blurb (enter in submission system): A short text summarizing in a single sentence the study (max. 200 characters including spaces). This text is used in conjunction with the titles of

papers, hence should be informative and complementary to the title. It should describe the context and significance of the findings for a general readership; it should be written in the present tense and refer to the work in the third person. Author names should not be mentioned.

B. MANUSCRIPT ORGANIZATION AND FORMATTING:

Sincerely,

Reviewer #1 (Comments to the Authors (Required)):

The authors have addressed all questions and concerns.

Reviewer #2 (Comments to the Authors (Required)):

The authors have addressed all my concerns and have done a great job addressing the other reviewer comments as well.

Reviewer #3 (Comments to the Authors (Required)):

I am happy with the response from the authors. I think that the manuscript is suitable for publication.

December 11, 2019

RE: Life Science Alliance Manuscript #LSA-2019-00476RR

Dr. Mafalda Escobar-Henriques
Institute for Genetics, Cologne Excellence Cluster on Cellular Stress Responses in Aging-Associated Diseases (CECAD), Center for Molecular Medicine Cologne (CMMC), University of Cologne
Joseph-Stelzmann-Straße 26
Cologne 50931
Germany

Dear Dr. Escobar-Henriques,

Thank you for submitting your Research Article entitled "Dual role of a GTPase conformational switch for membrane fusion by mitofusin ubiquitylation". It is a pleasure to let you know that your manuscript is now accepted for publication in Life Science Alliance. Congratulations on this interesting work.

DISTRIBUTION OF MATERIALS:

Again, congratulations on a very nice paper. I hope you found the review process to be constructive and are pleased with how the manuscript was handled editorially. We look forward to future exciting

submissions from your lab.

Sincerely,
